

# Assessment of source regions of the Zambezi River: implications for regional water security

Mauro Lourenco[1,2*], Rutledge S. Boyes[1,2], Fenton P.D. Cotterill[2], Tyrel Flügel[2], Frank C. Nyoni[3], Goabaone J. Ramatlapeng[1] and Rainer von Brandis[1,2]

[1] National Geographic Okavango Wilderness Project, Wild Bird Trust, Johannesburg, 2196, South Africa.
[2] The Wilderness Project Foundation 1050 Connecticut Ave, Suite 500, Washington D.C. 20036, United States of America
[3] Water Resources Management Authority (WARMA), Lusaka, Zambia

*Correspondence to*: Mauro Lourenco (mauro@wildbirdtrust.com)

**Abstract.** The Wilderness Project, in collaboration with the National Geographic Okavango Wilderness Project and the Wild Bird Trust, conducted the first known scientific on-river expeditions along the entire length of the Zambezi River—from its traditional source in Zambia and its most distal source in the Angolan Highlands Water Tower, all the way to the Indian Ocean. By combining ground measurement and high-resolution earth observation data, this study describes the overlooked contribution of Angolan Highlands Water Tower to the Zambezi River. Our findings show: 1. The Zambezi River length, measured from the Lungwebungu River, is 342 km longer (total length of 3,421 km to the Indian Ocean) than the traditional source, 2. We estimate that the Angolan portion of the Upper Zambezi and Lungwebungu sub-basins contributes 72.79% of the flow measured upstream of the Barotse Floodplain, 3. The Lungwebungu and Upper Zambezi sub-basins reduce elevated conductivity, salinity, and TDS, likely introduced by mining activities in the Kabompo sub-basin, and 4. The Upper Zambezi sub-basin has the highest proportion of wetland coverage, with 94.61% (19,184 km$^2$) in Angola (specifically the Kameia and Luena wetlands), promoting river health. This study highlights the recognition Angola deserves for its critical role in hosting the source and primary aquifer of the Zambezi River. Additionally, it underscores the need for improved monitoring and analysis of hydrological flows of the Zambezi Basin, particularly its Angolan tributaries. The Zambezi River Basin faces ongoing challenges from climate change, development, and human water use. Collaborative efforts among basin countries are crucial to balance development with environmental needs, safeguarding ecosystem services and the natural dynamics that sustain the Upper Zambezi's ecological integrity.

Key Words: Angola, Google Earth Engine, Water Tower, Zambia, Transboundary River

# 1 Introduction

Inherently dynamic landforms, rivers rank among the most renewable and sustainable sources of freshwater (Hanna et al., 2018). They provide a range of ecosystem services, including drinking water, sanitation, transportation, electricity generation, flood protection, and habitats for diverse species (Collins et al., 2024). Transboundary river basins are dynamic and complex,



featuring diverse landscapes and extensive temporal and spatial variability of resources (Lautze et al., 2017). The Zambezi River, the fourth longest in Africa, has a basin of approximately 1,370,000 km$^2$ that spans eight southern African countries: Angola, Botswana, Malawi, Mozambique, Namibia, Tanzania, Zimbabwe and Zambia (Moore et al., 2022). Recognised as one of Africa's most important river basins, the Zambezi is the focus of ongoing contested development, including water for
hydropower, agriculture, and the environment (Meier et al., 2011; Beilfuss, 2012; Arias et al., 2022). The basin possesses immense economic opportunity through its hydropower, which supports electricity demands; however, resource extraction, industrial and economic growth in the region are still in demand (Beilfuss, 2012; Arias et al., 2022). Climate change projections indicate a likely decline in water availability across the basin (Arias et al., 2022). In addition, the basin faces critical risks to the natural environment and ecosystem services, upon which the population, particularly rural communities, depend (Lautze
et al., 2017).

The Zambezi, meaning 'Great River' in ChiTonga, rises in a small spring near Kalene Hill in Zambia, and flows eastward for approximately 3,400 km before entering the Indian Ocean (Moore et al., 2022). The Zambezi Basin can be divided into three major segments: the Upper, Middle, and Lower Zambezi, each with distinct geomorphological characteristics (Wellington,
1955). The Upper Zambezi segment extends from the headwaters to Victoria Falls, one of the seven natural wonders of the world. The Middle Zambezi segment extends from below Victoria Falls and flows through the Gwembe and Mana Rift Grabens, delimited by the deep, narrow gorges incised by the river at Batoka, Kariba, Mupata and Cahora Bassa (Moore et al., 2009). The Lower Zambezi flows across the Mozambique coastal plain, extending from below Cahora Bassa Gorge to the Indian Ocean where its delta preserves a continuous, deep sedimentary record dating back to the Jurassic period (Moore et al.,
2022). The structure and organisation of the Zambezi River testifies to how repeated episodes of intracontinental rifting have reshaped the topology of its drainage. Yet remarkably, its modern landforms—primary channel and catchments—have evolved to the present day from recognisable ancient precursors. This means the extant topology of the Zambezi flows across ancient landforms inherited directly from a Proto-Zambezi river on the Gondwana Supercontinent; significant topological dynamics notwithstanding, this drainage network has persisted since the end of the Dwyka glaciation in Permian times (Moore et al.,
2009; 2022). Thus, the Zambezi is arguably the world's oldest river (Key et al. 2015; Moore et al., 2022).

The archaeological record preserved within the Zambezi Basin, particularly across the Zambian plateau and in the alluvial sediments above the Victoria Falls, attests to the importance of its wetland resources to human ancestors over the past 1 million years at least (Moore et al., 2022). The Zambezi River has a long history of recorded human settlement, reflecting the rich
heritage and culture of Khoi-San and Bantu peoples, the influence of Portuguese and British colonial rule, and more recent changes brought about by the construction of dams and bridges (Newitt, 2022). These include two of Africa's largest hydroelectric projects—the Kariba and Cahora Bassa Reservoirs—which have flooded vast rift valleys within the course of the Zambezi. An estimated 47 million people live in the basin, with 75% relying on subsistence agriculture and fisheries



(Cervigni et al., 2015). However, some regions in the basin remain sparsely populated, preserving some of the world's last
major remaining wilderness areas (Moore et al., 2022).

The Zambezi basin dominates Africa's Southern Savanna biome in the extent of its biotic habitats, particularly within the Zambezian phytochorion of angiosperm diversity. This is the largest of the phytochoria (regional domain of floristic diversity) in mainland Africa, as mapped by White (1983). In their congruence, the regional fauna, Lepidoptera, Odonata and vertebrates,
for example, exhibit a high congruence in diversity and endemism with the Zambezian vegetation (Frost et al., 2002; Timberlake and Chidumayo, 2011; Timberlake et al., 2018; Huntley, et al. 2019). The river's catchments and valleys support distinctive vegetation assemblages, exhibiting great structural diversity, which range from grassland dambos, floodplains and swamps (collectively mapped as the Zambezian Grasslands biome), to dry forests and thickets, and savanna woodlands. The Zambezian grasslands are the most widespread vegetation type in the region, such that it is difficult to find a catchment not
dominated by grassy dambos (White 1983; Cotterill, 2005). In particular, the distinctiveness of the diversity, endemism, and ecology of the Upper Zambezian fish fauna reflects the unique aquatic habitats of the Zambezi Okavango Miombo and Upper Zambezi Floodplain Ecoregions, redefined by Skelton (2024) as an update to the Freshwater Ecoregions of Africa mapped by Thieme et al. (2005).

Identifying and mapping a river's origin, particularly transboundary rivers, is important for several reasons, including water resource and river management, conservation, hydrological modelling and forecasting, water quality and pollution control, as well as policy, legal and political considerations. Compared to the well documented Zambian source regions of the Zambezi Basin (Meier et al., 2011; Beilfuss, 2012; Lautze et al., 2017; Hughes and Farinosi, 2020; Arias, 2022; Moore et al., 2022), the source waters originating in Angola are very understudied. The lack of field data from Angola is largely attributed to
widespread minefields, a legacy of the Angolan Civil War (1975–2002), which restricts access to remote locations (Conradie et al., 2016; Midgley and Engelbrecht, 2019), particularly in the headwaters of the Angolan Highlands Water Tower (AHWT: Lourenco et al., 2022). Moreover, hydrological assessments have primarily focused on the Middle and Lower Zambezi segments (Beilfuss, 2012; Lautze et al., 2017). Long-term studies also face significant data gaps due to the limited documentation of Angolan rivers, with most assessments relying on measurements along Zambian river courses (Moore et al.,
2022; Zambezi River Authority, 2024). Due to the lack of documentation regarding all of the Zambezi River's sources, the provenance of this regionally critical river can be disputed.

## 1.1 Defining river sources, multidisciplinary perspectives.

With respect to environmental policy, river sources are acknowledged through various legal mechanisms which recognise their significance to water resource management, environmental protection, and international cooperation (Giri, 2021). The United
Nations (UN) Watercourses Convention (1997) recognises the importance of river sources in the equitable and reasonable use of transboundary watercourses. It emphasises that the management of these watercourses should consider the river from source



to mouth (McCaffrey and Sinjela, 1998). In Southern Africa, the Southern African Development Community (SADC) Protocol on Shared Watercourses provides a legal framework for the cooperative management of transboundary water resources (SADC, 2000). Established in 2004, the Zambezi Watercourse Commission (ZAMCOM) Agreement adopts both the UN
Watercourses Convention and the SADC Protocol on Shared Watercourses (ZAMCOM, 2004). ZAMCOM aims to promote equitable and reasonable utilisation of the water resources of the Zambezi Watercourse, including efficient management and sustainable development by the member countries (ZAMCOM, 2004).

Formal geomorphological definitions of a river's source vary. This reflects the dynamic nature of rivers and the challenges
involved in identifying their precise origin(s). The headwater of a river is typically considered the farthest point on each of its tributaries, upstream of its mouth (Kammerer, 1987). The headwater is where surface runoff from rain, meltwater, and/or groundwater begins accumulating into a concentrated and consistent flow, forming a first-order tributary (Alexander et al., 2007). The tributary with the longest course downstream of the headwaters is the main stem (Kammerer, 1987). The United States Geological Survey (USGS) defines a river's length as either the distance from the mouth to the most distant headwater
source (irrespective of stream name), or from the mouth to the headwaters of the source stream (USGS, 2018). As most rivers have numerous tributaries, the longest tributary or stem is typically regarded as the source, regardless of its local name or usage (USGS, 2018).

A river source may be considered the most distant point from which water runs year-round (perennially: Messager et al., 2021),
or the furthest point from which water could flow ephemerally (Pastor et al., 2022). The latter definition includes seasonally-dry channels and suggests that the river's source waters could change monthly or yearly, depending on factors such as season, precipitation, climate cycles, or groundwater levels (Messager et al., 2021). Under these definitions, neither a lake (except those with no inflows) nor a confluence of tributaries can be considered a true river source, though both often serve as the starting point for the portion of the river that carries a single name (USGS, 2018). Nonetheless, the definition of a seasonal or
perennial river is contingent on the prevailing climatic regime, such that a seasonal river can contribute significant inflows in magnified pulses during periods of high rainfall. Moreover, when in flood, seasonal rivers not only deliver substantial water flow but also transport material fluxes that influence water chemistry. Importantly, a flowing seasonal river serves as a dynamic temporal resource for aquatic biodiversity, providing crucial spawning habitats for fishes and other aquatic biota (Minshull 2008, 2011). The seasonally dry floodplains surrounding the Upper Zambezi tributaries further exemplify the importance of
seasonal headwaters, as they experience large annual variations characteristic of tropical savanna climates (Beilfuss, 2012). Similar seasonal tributaries, particularly those on the south bank of the Middle Zambezi, contribute significant net inflows into the Kariba and Cahora Bassa reservoirs (Lautze et al., 2017; Arias et al., 2022).

In addition to water supply, water chemistry is a vital component of both river health and origin. Source waters strongly
influence water quality and flow, making them essential to water resource management (Alexander et al., 2007). Source water



regions provide unique habitats, support specialised species (Richardson, 2019), and contribute to water quality, flow conditions, and biodiversity. River sources play a critical role in maintaining overall river health and ecosystem services, particularly in regions where rivers support large human populations, such as the Zambezi (Arias et al., 2022), with the long-term sustainability of water quality being important for both human and environmental use. The topographical and hydrological importance of a stream in sustaining the river's flow and ecosystems should guide the definition of its source. Adopting multiple definitions ensures the true source is determined not solely by distance but also by the stream's contribution to the health of the river system.

River toponymy reflects the languages, cultures, and histories of the regions through which rivers flow (Mphasha et al., 2021). In addition to serving as geographical identifiers, naming systems encapsulate the socio-cultural narratives and ancestral heritage of local communities (Neethling, 2012). This diversity introduces challenges in defining a named river source due to several key factors: cultural and linguistic diversity, colonial influence and renaming, geopolitical tensions, historical and mythological significance, lack of documentation and standardisation, and the effects of modernisation and urbanisation (Neethling, 2012; Mphasha et al., 2021). Understanding and addressing these complexities is essential for effective environmental governance and preserving people's connection to their surroundings. This study acknowledges five distinct disciplines and their criteria for defining river sources (Table 1).



**Table 1. River source definition criteria across disciplines**

| Disciplines | Specific defining criterion for river sources | Examples | References and further reading |
|---|---|---|---|
| 1. Geographical criterion: The farthest point | The farthest point from the river mouth. The upstream terminus, the point at the uppermost end of the river's course from its mouth. The source is often near or on the edge of the watershed divide. | 1. The traditional source of the Zambezi River lies on the southern side of the Zambezi-Congo Watershed. 2. The source of the Colorado River is at the Continental Divide, separating the Atlantic and Pacific watersheds in North America. | Wohl and Merritts (2007); USGS (2018); Moore et al. (2022) |
| 2. Hydrological criterion: Water contribution | The primary water source that provides a consistent and significant volume of water, including seasonal variations in flow. Definitions acknowledge the furthest point from which water could possibly flow ephemerally. Multiple sources from a shared headwater source region could also be considered, in some cases the highest elevation point source within a shared headwater region can be considered as the source. | Since the mid-1600s, five separate river sources have been acknowledged for the Amazon River. These originate from the southwestern Peru headwater region. Of these, two main rivers have been considered as the main sources of the Amazon: the Marañón River has the higher flow rate, while the Mantaro River is the longest. | Lee (2014); Messager et al. (2021); Pastor et al. (2022) |
| 3. Environmental criterion: Origin of water and its contribution to the environment and people | Natural features from which the water originates and surfaces, including natural springs, glacial meltwater, wetlands and lakes. This also acknowledges the quality of water being contributed. Clean, unpolluted water may dilute pollutants and play a critical role in the river's ecological balance, including the ability to support critical habitats, species, biodiversity, ecosystem services, and freshwater provision for people. | 1. The source of the Lunguwebungu River (Angola) starts as a small trickle pouring out of a bowl-shaped peatland bog supported by groundwater. 2. The traditional source of the Zambezi River rises out of a forest spring in Zambia. 3. One of the sources of the Okavango River starts as a headwater lake at the origin of the Cuito River in Angola. 4. The Tees River (England) starts as a marshland. 5. The source of the Rhume River is the Rhume Spring in the Harz mountains of Germany. | Neal et al. (2008); Moore et al. (2022) |
| 4. Cultural and historical criterion | Traditional or historical sources based on cultural significance. Acknowledgement of local toponymy and nomenclature including cultural meanings attached to what is considered the river's source, which may not align with scientific or physical definitions. | The source of the Thames in England is traditionally defined according to the named river Thames rather than its longer tributary, the Churn, although not without contention. | Mphasha et al. (2021) |
| 5. Political and administrative criterion | Designates river sources officially for mapping, legal, and administrative purposes, potentially influenced by geographical, hydrological, or cultural defining criterion. | Lake Victoria is regarded as the main source of the Nile, specifically where the White Nile starts its journey. However, the most distant source is thought to be the Kagera River, a tributary of Lake Victoria, which has further tributaries extending into Rwanda and Burundi. | Dumont (2009); Tawfik (2016) |


The impending developments of the Zambezi's water resources highlights the importance of evidence-based decision-making, the river's headwater sources are a critical component to understanding and raising awareness of likely consequences. Ensuring sustainable development in this basin is crucial, as its hydrological landscape could change irreversibly in the next couple of



decades (Lautze et al., 2017). To support this, comprehensive scientific research is needed to better understand the river's

sources and its hydrological and ecological dynamics. In this context, The Wilderness Project (TWP), in collaboration with the National Geographic Okavango Wilderness Project (NGOWP) and the Wild Bird Trust (WBT), has completed the first on-river scientific research expeditions along the entire length of the Zambezi River. These river expeditions focused on both the traditional source of the Zambezi River in Zambia and the most distal source in the AHWT. Data collected include river topology (channel width and length), water chemistry, and discharge along the Zambezi River and its key tributaries.

Additional data on biodiversity (macroinvertebrate and fish assessments) and photographic surveys were collected but are not presented here.

Due to limited long-term monitoring infrastructure in the Upper Zambezi segment, particularly the Angolan portion, we supplement our empirical ground-measured data with several earth observation (EO) datasets to reevaluate the source of the

Zambezi River. Several topographical and hydrological characteristics are critical in maintaining natural river flows and ensuring ecosystem function, and each of these can be considered when defining river sources. In this research, we describe and quantify river distances, precipitation, flow accumulation, historical and empirical river discharge, seasonal flows, water quality, wetland inventories, and surface water dynamics. The lack of documentation on all of the Zambezi River sources, particularly those originating in Angola and the AHWT, is a key focus of this research. We include additional datasets that

have implications for transboundary river management within the Zambezi River Basin.

We hope the insights derived from this study motivate further exploration into quantifying African river sources in an empirical context. More accurate and comprehensive maps of river sources are increasingly essential for informing policy, including providing more inclusive and accurate geographical representations that honour both historical and contemporary geographical

reconstructions.

## 2 Materials and methods

### 2.1 Study Area

The Upper Zambezi segment (515,008 km$^2$) of the Zambezi Basin includes two of Africa's critical water towers: the Lufilian Arc and the AHWT (UNEP, 2010; Lourenco and Woodborne, 2023; Fig. 1). The Lufilian Arc is a belt of high ground along

the watershed between the Congo and Zambezi Basins. It gives rise to the traditional source of the Zambezi River, with headwater tributaries flowing southwards from the watershed divide (Moore et al., 2022). The western portion of the Upper Zambezi segment forms part of the AHWT, an important source region for major transboundary rivers: the Zambezi, Cunene, Okavango, and Congo Rivers, including Angola's most economically significant river, the Cuanza (Jackson, 1986; Lourenco and Woodborne, 2023). The basin's climate is largely controlled by the movement of the Inter-Tropical Convergence Zone

(ITCZ), with rainfall occurring predominantly during the November-to-March summer months, while the April-to-October





winter months are usually dry (Beilfuss, 2012). The natural flow regime of the Zambezi River reflects these rainfall patterns and is characterised by high seasonal and annual variability (Moore et al., 2022).



**Figure 1. The Zambezi River Basin, including the Upper Zambezi segment, water towers, Bulozi Graben (BG), Okavango Graben (OG), Machili Graben (MaG), Gwembe Graben (GG), Mana Rift Graben (MnG), and the CIFOR Tropical and Subtropical** 190 **Wetlands dataset from Gumbricht et al. (2017).**

These landscapes mantle the neotectonically active zone of the East African Rift Extension, centred on the Bulozi and Okavango-Machili Grabens (Pastier et al., 2017). Late Cenozoic uplift, along with subtle rifting, has reshaped the topology of the major tributaries, resulting in widespread drainage impedance. Structured by gentle relief, dambos—seasonally inundated 195 grasslands along drainage lines—characterise these densely vegetated tributaries and cover much of the African erosion surface. Across the Angola headwater tributaries, this region is classified as the Zambezi Okavango Miombo freshwater Ecoregion (Skelton, 2024). Their network extends across the deep Kalahari sediments that cover much of the Upper Zambezi segment, forming an archipelago of dambo wetlands and floodplains that impede runoff. Described as reservoir rivers by



Jackson (1986), the wetlands formed across this regional landscape include the Kameia, Lungwebungu, Luanginga, and

Cuando (Angola); the Liuwa and Barotse (Zambia); Linyanti (Botswana and Namibia); and Shesheke Flats (Namibia and

Zambia: Moore et al., 2012; 2022). These wetlands dominate the ecology of the Upper Zambezi Floodplains Ecoregion

(Skelton, 2024). Their vegetation consists of swamps and grasslands, classified as the Zambezian Grasslands biome by White

(1983). The unique biodiversity of these landscapes sustains important ecosystem services, maintained across the floodplains

and dambos that form a wetland archipelago, and hold a keystone function in the ecology and hydrology of the regional

landscape. Its overall distribution mantles the elevated Zambian and Angolan plateau, whose topography and drainage

topology have been reshaped by neotectonics across the East African Rift Extension (Cotterill, 2005; Moore et al., 2022).

## 2.2 Data

Empirical field-data collected within the Upper Zambezi segment includes discharge measurements at 47 sites and water

chemistry at 427 sites. Additionally, over 4,500 km of river was surveyed, from both Zambian and Angolan sources to the

Indian Ocean. Due to the lack of long-term hydrological and meteorological monitoring infrastructure, particularly within the

Angolan portion of the Upper Zambezi segment, and large-scale mapping requirement, we utilised several EO datasets to

supplement available empirical ground-based data to reevaluate the origin/source of the Zambezi River. We describe and

quantify river distance, precipitation, flow accumulation, historical discharge, seasonal flows, wetland inventories and surface

water dynamics using robust methodologies and EO data collected over long timescales (Table 2).




**Table 2. Datasets, data, sources and descriptions used in this study.**

| Dataset | Data source/software | Parameter Measured (unit) | Description |
|---|---|---|---|
| River flow path distance | 1. River length recorded using a Geode GNS3 receiver and GAIA GPS software on an iPhone13 ProMax. 2. WWF HydroRIVERS and Environmental Systems Research Institute's (ESRI) Trace Downstream tool. | River length (km) | 1. Continuous GPS track recording of river lines collected on river transect expeditions. 2. Determines the path water will take from a particular location to its furthest downhill path (ESRI, 2024). |
| Precipitation | CHIRPS precipitation data | Precipitation (mm.day$^{-1}$) | Climate Hazards Center InfraRed Precipitation with Station data (CHIRPS) is a 30+ year quasi-global rainfall dataset. CHIRPS incorporates 0.05° resolution satellite imagery with in-situ station data to create gridded rainfall timeseries (Funk et al., 2015). |
| Flow accumulation | WWF HydroSHEDS Flow Accumulation data | Flow Accumulation (no. cells) | This dataset defines the amount of upstream area (in number of cells) draining into each cell (Lehner et al., 2008). |
| Empirical river discharge | 1. Acoustic Doppler Current Profiler (ADCP) to measure water discharge at key sites. 2. Historical hydrological data from gauge stations maintained and operated by the Water Resources Management Authority (WARMA) of Zambia. | 1. Mean discharge (m$^3$.s$^{-1}$) 2. Mean monthly discharge (m$^3$.s$^{-1}$) | 1. SonTek RS5 and M9 Acoustic Doppler Current Profiler (ADCP) to measure water discharge at key locations on rivers within the Upper Zambezi segment. We pulled the ADCP across the channel, while remaining within acceptable limits in terms of sampling speed and trajectory. At each site, we conducted 4 transects across the river so that variance could be calculated. In cases where variance was too high (COV > 0,5), the site was resampled. 2. Historical gauge station data provided by WARMA in Zambia, mean monthly discharge data for those same stations also quoted from various sources in the literature. |
| Modelled river discharge | 1. Globally gauge-corrected monthly river flow. 2. Annual natural river discharge dataset for Africa. | 1. Mean monthly discharge (m$^3$.s$^{-1}$) 2. Annual discharge (km$^3$) | 1. The mean discharge value for individual months over the timeseries between January 1980 and December 2009 from Collins et al. (2024). 2. Discharge values in cubic kilometres on an annual basis from 2001 to 2021 from Akpoti et al. (2024). |
| Wetland inventory | CIFOR Tropical and Subtropical Wetlands Distribution V3 | Wetland occurrence | The geographical distribution of wetlands in tropical and subtropical areas by the Centre for International Forestry Research (CIFOR) from Gumbricht et al. (2017). |
| Empirical water chemistry | In-Situ Aqua TROLL 600 multi-parameter sonde | Temperature (°C), pH (pH), specific conductivity (µS/cm), salinity (PSU), total dissolved solids (TDS) (ppt), dissolved oxygen (DO) (mg/L) and turbidity (NTU) | Aqua TROLL 600 multi-parameter sondes were used to measure water temperature, pH , specific conductivity, salinity, TDS and DO in the river water. |
| Surface water inventory | JRC global surface water inventory | Water occurrence (frequency) and water seasonality (months) | Joint Research Centre (JRC) global surface water inventory dataset maps the location and temporal distribution of surface water from 1984 to 2021 and provides statistics on the extent and change of those water surfaces (Pekel et al., 2016). |
| Lake/reservoir inventory | WWF HydroLAKES | Lakes and reservoirs occurrence (km$^2$) | HydroLAKES provides the shoreline polygons of all global lakes with a surface area of at least 0.1 km$^2$ (Messager et al., 2016). |



## 2.3 Distance of tributaries

The traditional source of the Zambezi River begins in northwest Zambia from a small spring on the Congo-Zambezi watershed that separates the river from northwest-flowing tributaries of the Congo (Moore et al., 2022). For approximately 30 km, the
Zambezi headwaters flow north toward the Congo Basin before changing course to the southwest, a direction it maintains for a further 200 km into Angola. The river then turns south-eastwards toward the Indian Ocean (Moore et al., 2022). However, additional tributaries of the Zambezi exist within other major sub-basins (Fig. 2).

Map showing the Zambezi River Basin with labelled sub-basins and tributaries.

Legend:
- Country Boundary
- Zambezi River Basin
- Upper Zambezi
- Lungwebungu
- Kabompo
- Luanginga
- Cuando Chobe
- Barotse
- Kafue
- Karia
- Mupata
- Luangwa
- Tete
- Shire
- Zambezi Delta

**Figure 2. The river names and flow paths within each HydroSHEDS level 04 sub-basin of the Zambezi Basin were measured using**
**ESRI's Trace Downstream Tool. Note the distinction between the Upper Zambezi segment—which divides the Zambezi River Basin into Upper, Middle, and Lower segments—and the Upper Zambezi sub-basin—the sub-basin where the traditional source of the Zambezi River begins.**




A complete GNSS track of the traditional source of the Zambezi River and the most distal source in Angola were recorded along the physical river channel during individual TWP river expeditions using a Geode GNS3 receiver and GAIA GPS software on an iPhone13 Pro Max. These measurements were taken from the sources of these two rivers to their confluence near the town of Lukulu (14.4076° S, 23.2588° E) in Zambia, upstream of the Barotse Floodplain. Following the completion of the TWP river expeditions, which traversed the entire length of the Zambezi River to the Indian Ocean, the full river length was calculated from the expedition track data. This provides an accurate field-validated measurement of the Zambezi's total length, from both its traditional source in Zambia and its most distal source in Angola. Note, for both Kariba and Cahora Bassa Reservoirs, the straight-line navigable length across the lakes was used, although the expedition teams surveyed the southern shorelines.

The Environmental Systems Research Institute (ESRI) Trace Downstream Tool was used to determine the overall length of the additional major tributaries within the Zambezi basin (ESRI, 2024). The Trace Downstream tool uses a hosted digital elevation model (DEM)—in this region, the 90 m HydroSHEDS DEM—to trace downstream flow paths from a particular location to its furthest downhill path/the ocean (ESRI, 2024). The source of each river tributary was determined using high-resolution satellite imagery and the African HydroRIVERS data (Lehner and Grill, 2013). Fourteen river lengths were measured across the Zambezi Basin to the river mouth. The highest stream order tributary/tributaries according to the HydroRIVERS dataset within each HydroSHEDS level 04 sub-basin (Linke et al., 2019) were measured, except for the Zambezi Delta sub-basin near the river mouth. Elevation from NASA's Shuttle Radar Topography Mission (SRTM) 30 m DEM was extracted at 10 km intervals along each river line in ArcGIS Pro. Given the 90 m DEM input, the ESRI Trace Downstream Tool likely underestimates river length, because not all river meanders are captured. To address this limitation, the longest flow path line (Lungwebungu) identified by the tool and the traditional Zambezi source were digitised manually along the river's centre using high-resolution satellite imagery in Google Earth Engine (GEE). The river lengths measured include: 1) recorded track length (TL), 2) ESRI Trace Downstream length (EL) and 3) digitised length (DL). The straight-line distance (l), EL error percentage, and river sinuosity were also measured.

## 2.4 Precipitation

A large portion of the inhabitants of the Zambezi River Basin depend on its water resources (Fant et al., 2015). However, the basin frequently experiences extreme floods and droughts, and climate change is expected to exacerbate these unfavourable climatological and hydrological conditions (Hughes and Farinosi, 2020; Dube and Nhamo, 2023). There is no robust network of meteorological weather stations across the Zambezi Basin; therefore, precipitation data for this study were obtained from the Climate Hazards Center InfraRed Precipitation with Station data (CHIRPS) 30+ year quasi-global rainfall dataset (Funk et al., 2015). CHIRPS synthesises 0.05° resolution (~5 km) satellite imagery with in situ station data to create a gridded rainfall time series (Funk et al., 2015). These summarised data describe the seasonal and spatial distribution of precipitation over the basin, but they are not utilised to derive higher-order climatic trend analyses.



The mean precipitation (mm/day) for each month across the 43-year time series from 1981-01-01 to 2023-12-31 was calculated for each individual HydroSHEDS level 04 sub-basin at 5 km resolution (Fig. 3a) in GEE. The CHIRPS data indicate strong seasonality in precipitation across the Zambezi Basin. Mean values per day were calculated to avoid bias in months with fewer days. According to the CHIRPS data, rainfall begins in September each year, peaking in January (mean across all sub-basins

= 7.24 mm/day). The wet season lasts through to April, followed by an almost entirely dry season from May to August, with August being the driest month (mean across all sub-basins = 0.06 mm/day). The Zambezi Delta sub-basin in the southeast remains the wettest during the dry season compared to the other sub-basins.

Mean daily precipitation, represented spatially across the Zambezi Basin, is characterised by a latitudinal gradient, with higher

daily means in the northern sub-basins and lower daily means in the southern sub-basins (Fig. 3b). Precipitation is also higher over the three water towers and in the east nearer the Indian Ocean, while the southwest of the Zambezi Basin remains comparatively dry. Although the traditional source of the Zambezi begins in the northwestern corner of Zambia, the river enters Angola approximately 89 km (from recorded track data) downriver. The majority (86.77%) of the Upper Zambezi sub-basin lies within Angola, where much of the precipitation occurs and where the Luena, Luateche, and Chifumage Rivers

originate.

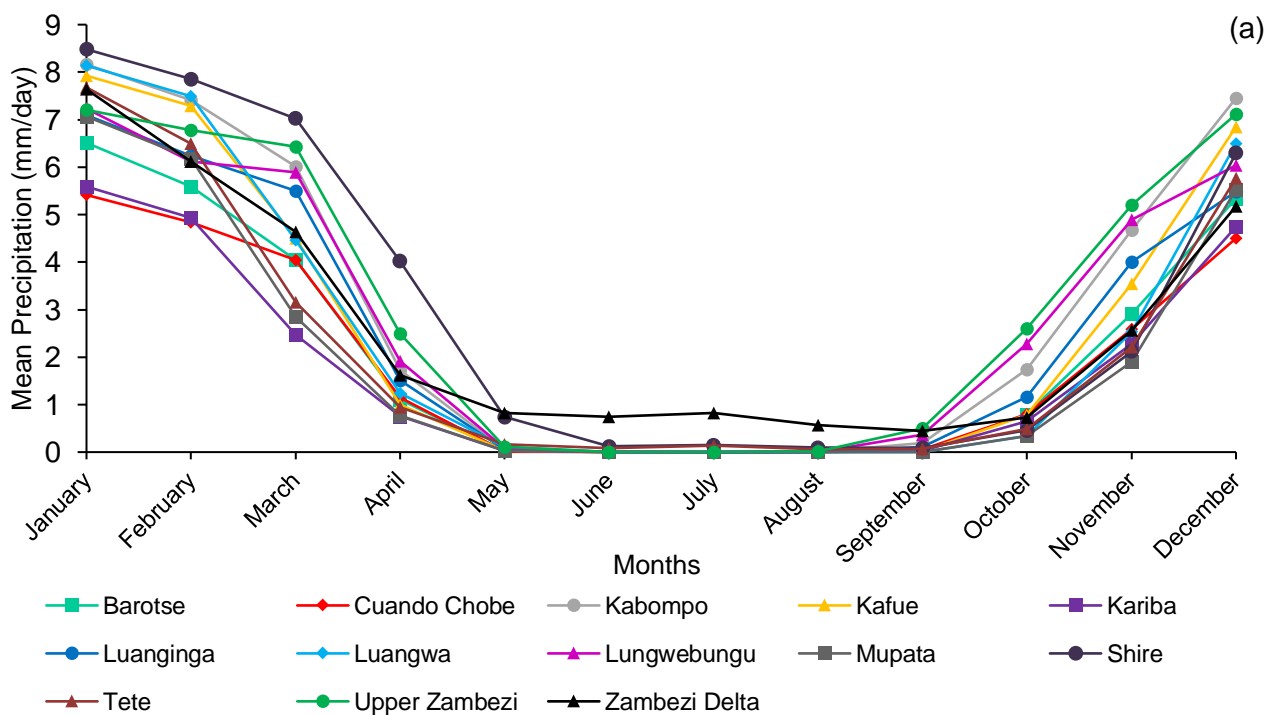





**Figure 3. (a) The mean daily precipitation per month for each HydroSHEDS level 04 sub-basin. (b) The mean daily CHIRPS precipitation (Funk et al., 2015) across the Zambezi Basin indicating that the northern basin has a higher daily precipitation compared to the south.**

## 2.5 River discharge and flow accumulation

On river expeditions, we used a SonTek RS5 and M9 Acoustic Doppler Current Profiler (ADCP) to measure water discharge along the main stem of the Zambezi River and at major confluences. To obtain a discharge measurement, the ADCP was pulled across the river channel while remaining within acceptable limits in terms of sampling speed and trajectory. At each site, we conducted four transect measurements to calculate variance. In cases where the coefficient of variation was too high (COV > 0.05), resampling was required. As our ADCP readings provide a single discharge measurement at a point in time, we report the discharge values and the date of recording, as seasonality is an important consideration in understanding temporal variations in flow. These data cannot be used to determine higher-order discharge trends or analyses. Instead, we combined our empirical ADCP discharge measurements with historical gauge station data, the WWF HydroSHEDS Flow Accumulation dataset, and



the global and regional river discharge datasets. This approach allows for a relative flow comparison of rivers in the HydroSHEDS level 04 sub-basin and is used to determine which rivers contribute the most to overall flow within the Upper
Zambezi segment.

The Victoria Falls hydrological station lies at the base of the unregulated Upper Zambezi segment, and has been measuring river discharge since 1907 (Moore et al., 2022). Presently, the Zambezi River Authority monitors daily river flows at thirteen stations, with data from four stations publicly available, albeit this access is restricted to the last two weeks only; historical
data are not available on their website (Zambezi River Authority, 2024). Nevertheless, we have obtained historical gauge station data for some of these stations through the Zambian Water Resources Management Authority (WARMA) and various references in the literature. In addition, there are no hydrological monitoring stations for rivers within the Angolan portion of the Upper Zambezi segment (Schleiss et al., 2017). To fill this data gap, EO datasets are used to infer both flow accumulation and modelled river discharge. We use the flow accumulation dataset derived from HydroSHEDS, which defines the amount
of upstream area (in number of cells) draining into each cell. The drainage direction layer defines which cells flow into the target cell. The number of accumulated cells serves as a measure of the upstream catchment area, with values ranging from 1 at topographic highs (river sources) to very large numbers (on the order of millions of cells) at the mouths of large rivers (Lehner et al., 2008). We calculated the number of accumulated cells within each HydroSHEDS level 04 sub-basin to estimate the proportion of flow accumulation within the Zambezi Basin.


Several investigations into modelling the hydrological flow regimes and discharge of specific sections within the Upper Zambezi segment have been conducted (Meier et al., 2011; Zimba et al., 2018; Hughes et al., 2020; Hughes et al., 2023). While modelled river discharge data are subject to inherent uncertainties, the large wetlands, underlying Kalahari sands, and relatively poor understanding of water exchange dynamics between the river channels and wetland storage areas of the Upper Zambezi
segment further compound this issue (Hughes et al., 2020). To investigate whether global and regional datasets can help address these challenges, we use a globally gauge-corrected monthly river flow and storage dataset from Collins et al. (2024) and an annual natural river discharge dataset for Africa from Akpoti et al. (2024). These datasets were compared to historical gauge station data for selected stations within the Upper Zambezi segment. The Collins et al. (2024) dataset provides mean discharge values in cubic meters per second ($m^3.s^{-1}$) for individual months over the time series from January 1980 to December
2009. These mean discharge values were averaged across all months to calculate an average discharge per month over a calendar year. The Akpoti et al. (2024) dataset provides discharge values in cubic kilometres on an annual basis from 2001 to 2021; these values were converted to mean annual discharge values in $m^3.s^{-1}$.

**2.6 Water chemistry**

Source waters have a strong influence on river water quality and are an essential component of water resource management
and overall river health (Alexander et al., 2007). However, water chemistry data within the Upper Zambezi segment are largely



unavailable, and where available, the data are insufficiently sampled over extended periods to identify contamination sources, conduct trend analyses, or establish correlations. While water chemistry measurements were conducted during TWP expeditions within the Upper Zambezi segment, these measurements are limited to single data points sampled along major tributaries and cannot yet enable comprehensive analysis without repeated measurement and consideration of seasonal

variations. The Upper Zambezi segment is characterised by numerous streams and extensive wetlands and floodplains, in contrast to the more hydrologically regulated Middle and Lower Zambezi segments, which feature large-scale reservoirs and lakes (Beilfuss, 2012; Schleiss et al., 2017). Hydropower development and land-use changes have been evaluated for their influence on surface water chemistry in the Middle Zambezi segment (Winton et al., 2021), but no study has focused on water chemistry across the entire basin. Given the limited availability of comprehensive water chemistry data across the Zambezi

Basin, we have incorporated a wetland and surface water inventory for each HydroSHEDS Level 04 sub-basin to complement existing data and serve as a supporting proxy indicator of overall river and ecosystem health.

Water chemistry parameters were measured along different tributaries of the Upper Zambezi segment using an In-Situ Aqua TROLL 600 multi-parameter sonde. The parameters include river temperature (°C), pH (pH), specific conductivity (µS/cm),

salinity (PSU), total dissolved solids (TDS) (ppt), dissolved oxygen (DO) (mg/L), and turbidity (NTU). The rivers and tributaries measured include the Cuando (first half in 2018, second half in 2023), Kembo (2018), Lungwebungu (Angolan section in 2022, Zambian section in 2023), and the traditional source of the Zambezi (2023) along the main river stem to Livingstone, near Victoria Falls. A total of 427 sites were measured across the lengths of these rivers, with measurements conducted at 10 km intervals along the river. During field measurements, the sonde was submerged for at least 30 seconds to

ensure stable readings. Data were recorded for 100 seconds, with one reading every 2 seconds, and the stabilised median value of these was selected for each water quality parameter. Sonde sensors were calibrated and replaced according to the manufacturer's recommended frequency to maintain accuracy (In-Situ Inc, 2024). The locations of each of these water quality sites were mapped using ArcGIS Pro.

## 2.7 Wetland and surface water inventory

There are multiple studies describing the individual major wetlands within the Zambezi Basin (Timberlake, 2000; Seyam et al., 2001; Beilfuss, 2012; Schleiss et al., 2017; Moore et al., 2022); however, with the partial exception of White (1983), there is no documented quantification and classification of wetland extent across the entire Zambezi Basin. We use the Centre for International Forestry Research (CIFOR) Tropical and Subtropical Wetlands Distribution V3 dataset from Gumbricht et al. (2017), with a mapped resolution of 231 m. The CIFOR dataset employed a hydrological model combined with an annual time

series of satellite-derived estimations of soil moisture, reflecting the dynamics of water movement and the extent of surface wetness. These were then combined with geomorphological data to provide an extensive and accurate representation of wetland ecosystems, see Gumbricht et al. (2017) for detailed methodology. This wetland raster dataset was clipped to the individual HydroSHEDS level 04 sub-basins within the Zambezi Basin, and the total extent of each wetland class, along with the



percentage cover of wetlands for each basin, was quantified by summing each CIFOR wetland class, excluding the *Open Water*
class.

The Joint Research Centre (JRC) global surface water inventory (Pekel et al., 2016) was used to quantify the water occurrence
and seasonality of each HydroSHEDS level 04 sub-basin within the Zambezi River Basin. This dataset contains the location
and temporal distribution of surface water from 1984 to 2021, and provides statistics on the extent and change of those water
surfaces (Pekel et al., 2016). The water occurrence (frequency with which water was present from: 0 = never present to 100 =
permanently present) and seasonality (mean number of months water is present within a calendar year) were extracted for the
entire 38-year image acquisition period (Pekel et al., 2016). Within each sub-basin, the percentage of individual pixels that
indicated the presence of water were placed within distribution tables for the occurrence band (1-100: frequency) and
seasonality band (1-12: months) to describe the surface water dynamics within each individual HydroSHEDS level 04 sub-
basin in the Zambezi River Basin over the entire JRC time series. In addition to the JRC global surface water inventory, we
used the HydroLAKES dataset, which provides the shoreline polygons of all global lakes and reservoirs with a surface area of
at least 0.01 km$^2$ (Messager et al., 2016). This product was developed using a suite of auxiliary data sources of lake polygons
and gridded lake surface areas. The dataset was clipped to each individual HydroSHEDS level 04 sub-basin and administrative
country boundary within the Zambezi River Basin in ArcGIS Pro. All maps presented were produced using ArcGIS Pro
software.

## 3 Results

### 3.1 Distance of tributaries

Following the completion of the TWP Zambezi River expeditions, the length of the entire Zambezi River to the Indian Ocean
was calculated using expedition track data (Table 3). The total length from the Lungwebungu source to the Indian Ocean is
3,421 km, whereas the total length from the traditional Zambezi source to the Indian Ocean is 3,079 km. Notably, the lengths
across the two major reservoirs included the straight-line navigable routes across the water bodies; the TL were not used as
they followed the lake's southern shorelines, which would overestimate the river's true length.





**Table 3. River and straight-line lengths of the traditional Zambezi source and Lungwebungu River, with a colour palette representing the EL error percentage.**

| River Name | Expedition Track length (TL) (km) | Digitised River length (DL) (km) | ESRI Trace downstream length (EL) (km) | EL Error (%) | Straight line length (l) (km) | River Sinuosity (TL/l) |
|---|---|---|---|---|---|---|
| Traditional Zambezi Source to Lukulu | 747 | 755 | 653 | -12,58% | 347 | 2.15 |
| Lungwebungu source to Lukulu | 1,089 | 1,098 | 819 | -24,79% | 557 | 1.95 |
| Traditional Zambezi Source to Indian Ocean | 3,079 | - | 3,051 | -0,91% | 1,535 | 2.01 |
| Lungwebungu source to Indian Ocean | 3,421 | - | 3,217 | -5,72% | 2,045 | 1.67 |

Of the 14 calculated river lengths (ESRI Trace Downstream Tool), the three longest rivers are: 1) Lungwebungu (3,217 km), 2) Luateche (3,169 km), and 3) West Lunga (3,122 km) (Fig. 4). The traditional source of the Zambezi ranks 5[th] with a length of 3,051 km, 166 km shorter than the Lungwebungu River according to this tool. Of the five longest river lines, three originate in Angola (Lungwebungu, Luateche and Luena: 3,114 km) while two originate in Zambia (West Lunga and Zambezi). Notably the Manyame River (1,387 km), originating in Zimbabwe, starts at the highest elevation (1,660 masl) but its source lies within a separate headwater region.

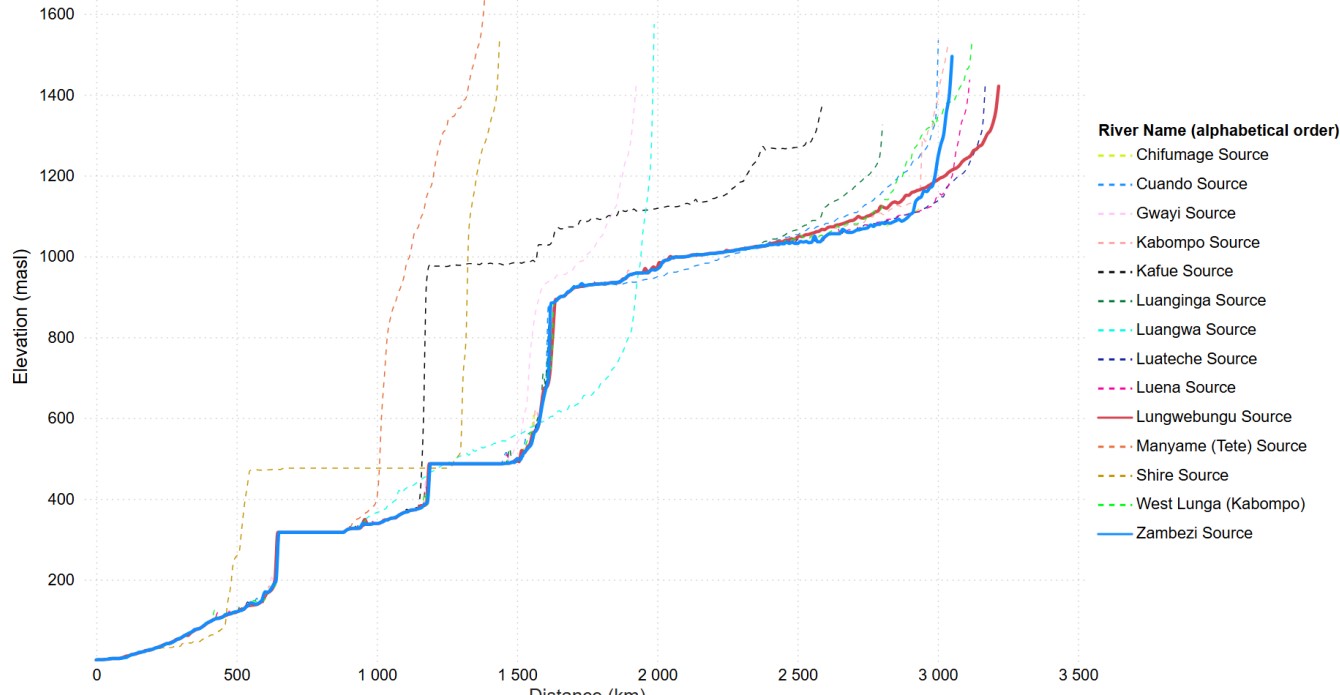

**Figure 4. Longitudinal profiles of Zambezi River Basin tributaries to the river mouth, derived using ESRI's Trace Downstream Tool. Steep vertical drops indicate large waterfalls, gorges and reservoir dam walls, while flat horizontal lines indicate large lakes or reservoirs.**



The comparison of ESRI Trace Downstream Tool lengths (EL) with the measured expedition track lengths (TL) and manually digitised lengths (DL) indicates that the Lungwebungu is approximately 342 km longer than the traditional source of the Zambezi. However, the river sinuosity is higher for the traditional Zambezi River source (2.15) compared to the Lungwebungu River (1.95) up to the confluence at Lukulu (Table 3). In addition, the TL show a -12.58% error for the traditional Zambezi source and a -24.79% error for the Lungwebungu River compared to the EL up to Lukulu. This tool should be used with caution for rivers with high sinuosity.

### 3.2 Flow accumulation and river discharge

According to flow accumulation, the Upper Zambezi segment has the highest percentage (37.50%) of accumulated cells compared to the middle and lower segments (Table 4). Within the Upper Zambezi, the Cuando/Chobe sub-basin provides 11.35% of the total number of accumulated cells as a measure of the upstream catchment area. The Barotse sub-basin contributes 7.79% of accumulated cells; however, this basin begins below several other source basins, including the Upper Zambezi, Lungwebungu, Kabompo, and Luanginga. The Tete sub-basin, part of the Lower Zambezi segment, has the highest flow accumulation contribution (14.40%) and spans across parts of Mozambique, Zimbabwe, and Zambia.

**Table 4. The WWF HydroSHEDS Flow Accumulation for each HydroSHEDS level 04 sub-basin within the Zambezi River Basin.**

| HydroSHEDS level 04 Sub-basin | Number of accumulated cells | Proportion (%) |
|---|---:|---:|
| UPPER ZAMBEZI SEGMENT TOTAL | 2,493,806 | 37.50% |
| Upper Zambezi | 447,043 | 6.72% |
| Lungwebungu | 224,961 | 3.38% |
| Kabompo | 346,830 | 5.22% |
| Luanginga | 202,529 | 3.05% |
| Cuando/Chobe | 754,660 | 11.35% |
| Barotse | 517,783 | 7.79% |
| MIDDLE ZAMBEZI SEGMENT TOTAL | 2,373,253 | 35.69% |
| Kariba | 812,330 | 12.22% |
| Kafue | 753,137 | 11.33% |
| Mupata | 92,307 | 1.39% |
| Luangwa | 715,479 | 10.76% |
| LOWER ZAMBEZI SEGMENT TOTAL | 1,782,323 | 26.80% |
| Tete | 957,620 | 14.40% |
| Shire | 769,778 | 11.58% |
| Zambezi Delta | 54,925 | 0.83% |
| ZAMBEZI RIVER BASIN TOTAL | 6,649,382 | 100.00% |





Within the Upper Zambezi Segment, historical discharge data were obtained from eight separate gauge stations. The globally gauge-corrected monthly river flow and storage dataset from Collins et al. (2024) and an annual natural river discharge dataset

for Africa from Akpoti et al. (2024) were compared with this historical gauged station data (Table 5). The locations of each station were mapped (see Supporting Material, Fig. S1). It is evident that both the Collins et al. (2024) and Akpoti et al. (2024) datasets diverge greatly from the historical gauged station data within this geographical setting. Note that the Collins et al. (2024) dataset provides monthly mean discharge values, and it diverges the most for the station along the Cuando River (at the Kongola gauge station), where discharge is inflated (modelled peak flow of 1,845 $m^3.s^{-1}$ vs gauged flow of 37 $m^3.s^{-1}$) and

shown to occur from June to August, not in February. The Akpoti et al. (2024) dataset provides an annual discharge that has been averaged over the period 2001-2021; this dataset also diverges greatly at the Kongola gauge station (modelled annual flow of 237 $m^3.s^{-1}$ vs gauged flow of 33 $m^3.s^{-1}$). See Supporting Material, Fig. S2, which includes the monthly mean flows for each of the eight historical gauge stations. Considering that direct extraction of discharge flows for each major tributary from Collins et al. (2024) and Akpoti et al. (2024) would not reveal any meaningful results with certainty, we make use of the

historical gauge data, the WWF HydroSHEDS Flow Accumulation data and our empirical ADCP discharge to determine which of these rivers have the highest mean monthly discharge and overall flow within the Upper Zambezi segment.

**Table 5. Mean monthly measured and modelled flows ($m^3.s^{-1}$) for historical gauge station locations within the Upper Zambezi segment.**

| Hydro-Station | Chavuma station[1] | Chavuma model[2(3)] | Watopa station* | Watopa model[2(3)] | Lukulu station[1] | Lukulu model[2(3)] | Kalabo station* | Kalabo model[2(3)] | Senanga station[4] | Senanga model[2(3)] | Katima Mulilo station[1] | Katima Mulilo model[2(3)] | Kongola station[5] | Kongola model[2(3)] | Vic Falls station[1] | Vic Falls model[2(3)] |
|---|---|---|---|---|---|---|---|---|---|---|---|---|---|---|---|---|
| Period (years) | 1959–2002 | 1980–2009 | 1958–2024 | 1980–2009 | 1950–2002 | 1980–2009 | 1961–2022 | 1980–2009 | 1947–1993 | 1980–2009 | 1967–2002 | 1980–2009 | 1970–2020 | 1980–2009 | 1951–2002 | 1980–2009 |
| Jan | 655 | 638 | 305 | 644 | 803 | 1,743 | 38 | 149 | 793 | 2,113 | 678 | 2,281 | 29 | 1,041 | 686 | 3,780 |
| Feb | 1,411 | 876 | 474 | 736 | 1,294 | 2,282 | 66 | 276 | 1,307 | 2,916 | 1,211 | 3,190 | 32 | 1,845 | 1,184 | 5,777 |
| Mar | 2,031 | 875 | 623 | 630 | 1,645 | 2,126 | 91 | 266 | 2,087 | 2,686 | 2,374 | 2,921 | 35 | 1,664 | 2,175 | 5,129 |
| Apr | 1,770 | 433 | 526 | 239 | 1,523 | 922 | 104 | 95 | 2,319 | 1,124 | 3,129 | 1,212 | 35 | 631 | 3,007 | 2,028 |
| May | 684 | 124 | 236 | 61 | 944 | 248 | 100 | 25 | 1,946 | 306 | 2,427 | 336 | 34 | 184 | 2,613 | 584 |
| Jun | 310 | 50 | 148 | 25 | 575 | 97 | 75 | 11 | 1,280 | 125 | 1,326 | 141 | 37 | 88 | 1,621 | 265 |
| Jul | 188 | 30 | 122 | 15 | 434 | 56 | 50 | 7 | 745 | 74 | 691 | 84 | 37 | 57 | 845 | 167 |
| Aug | 124 | 21 | 101 | 10 | 361 | 40 | 31 | 5 | 492 | 52 | 467 | 60 | 37 | 43 | 519 | 123 |
| Sep | 83 | 27 | 79 | 14 | 306 | 54 | 19 | 5 | 374 | 66 | 364 | 73 | 35 | 40 | 376 | 131 |
| Oct | 68 | 70 | 68 | 70 | 271 | 184 | 13 | 12 | 310 | 218 | 306 | 236 | 32 | 101 | 293 | 388 |
| Nov | 94 | 184 | 87 | 179 | 310 | 486 | 13 | 44 | 340 | 586 | 320 | 633 | 28 | 330 | 297 | 1,106 |
| Dec | 228 | 349 | 165 | 352 | 468 | 943 | 18 | 85 | 487 | 1,146 | 430 | 1,257 | 27 | 630 | 438 | 2,241 |
| **Mean annual** | **637** | **307 (244)** | **244** | **248 (216)** | **745** | **765 (606)** | **52** | **82 (94)** | **1,040** | **951 (812)** | **1,144** | **1,035 (935)** | **33** | **554 (237)** | **1,171** | **1,810 (1,271)** |

Notes for Table 5: [1]Moore et al. (2022), [2]Collins et al., (2024), [3]Akpoti et al. (2024), [4]Hughes et al. (2020), [5]Pallett et al., (2022). The

Akpoti et al. (2024) dataset includes only the mean annual discharge for each station location over the period 2001–2021. *The Watopa Pontoon and Kalabo station data was provided by WARMA.



The Chavuma station (Fig. 5) is located in Zambia, with 381,670 flow accumulation pixels, representing 85.38% of the total for the Upper Zambezi sub-basin, and has a mean annual discharge of 637 m$^3$.s$^{-1}$. Further downstream, the Watopa Pontoon

station includes two major tributaries—the Kabompo and West Lunga—has 318,779 flow accumulation pixels, representing 91.91% of the Kabompo sub-basin, and a mean annual discharge of 244 m$^3$.s$^{-1}$. The Lukulu station (Fig. 5) is located just upstream of the Barotse Floodplain, with 1,019,063 flow accumulation pixels. Its flows originate from a combination of the Upper Zambezi, Lungwebungu, and Kabompo sub-basins. Despite these two additional tributaries, the mean annual flow is 745 m$^3$.s$^{-1}$, only 108 m$^3$.s$^{-1}$ more than at Chavuma. Historical station data show that during the peak flow months of February,

March, and April, Chavuma has a higher mean monthly discharge than Lukulu, whereas the drier months of May to October show the opposite. At Sananga, the gauge station is located near the southern end of the Barotse Floodplain, with 1,386,052 flow accumulation pixels,



incorporating flows from the Luanginga sub-basin. At Kalabo station which has 159,284 flow accumulation pixels representing 78.64% of the Luanginga sub-basin, the mean annual flow is 52 $m^3.s^{-1}$. Although the flow regime is similar between Sananga

and Lukulu, the mean annual flow at Sananga is 1,040 $m^3.s^{-1}$, 295 $m^3.s^{-1}$ greater than at Lukulu, due primarily to additional inputs from the Barotse, Liuwa Plains, and Luanginga wetlands and tributaries.

At Katima Mulilo, 1,611,429 accumulated pixels, mean annual flows are 1,144 $m^3.s^{-1}$, 104 $m^3.s^{-1}$ greater than at Sananga, with inputs from tributaries within the eastern section of the Barotse sub-basin. Mean annual flows at the base of the Upper

Zambezi segment at Victoria Falls increase to just 1,171 $m^3.s^{-1}$, despite contributions from the Cuando Chobe sub-basin. The Cuando River flows measured at Kongola, 567,376 flow accumulation pixels, indicates a mean annual flow of just 33 $m^3.s^{-1}$ despite being the sub-basin with the greatest number of accumulated cells. The Upper Zambezi is the largest contributing sub-basin in terms of mean annual river discharge, among the longest tributaries, according to selected locations with sufficient historical gauge data. The flow accumulation dataset is used to approximate contributions from each, with the number of flow

accumulation pixels in descending order: Luena and Luateche (114,727); Traditional Zambezi Source (106,346); and Chifumage (89,880).

Selected empirical discharge data using an ADCP, collected at 47 separate locations within the Upper Zambezi segment, are listed in their chronological order of sampling (Fig. 5). Notable measured discharge estimates include: the Lungwebungu River

(site 16: 335.70 $m^3.s^{-1}$ on 2023/03/10), contributing approximately 22.82% of the 1,471.10 $m^3.s^{-1}$ measured along the Zambezi at site 17 on the same day. Discharge measured at the Zambezi River (site 25, just downstream of Chavuma) was 928.56 $m^3.s^{-1}$ on 2023/05/17, which is comparable to the average discharge at the hydrological station between April (1,770 $m^3.s^{-1}$) and May (684 $m^3.s^{-1}$). In summary, discharge measured at the end of the Kabompo River (site 26: 318.08 $m^3.s^{-1}$ on 2023/05/21) and Lungwebungu River (site 27: 193.85 $m^3.s^{-1}$), along with measurements on the Zambezi River upstream of the

Lungwebungu River (site 28: 1,194.19 $m^3.s^{-1}$ on 2023/05/22), indicates that the Kabompo and Lungwebungu Rivers were contributing approximately 22.92% and 13.97%, respectively, at the time of measurement.

Discharge measured along the Zambezi River at site 33 was 28.52 $m^3.s^{-1}$ on 2023/07/13. This increased downstream due to several inflows and tributaries, including the Luisaba River (site 35: 37.72 $m^3.s^{-1}$ on 2023/07/16), reaching 105.95 $m^3.s^{-1}$ at

site 36 on 2023/07/18. The inflow of the Chifumage River (site 38: 21.17 $m^3.s^{-1}$ on 2023/07/21) increased the flow to 135.41 $m^3.s^{-1}$ on 2023/07/22, measured upstream of the Luena River tributary. The Luena River (site 39: 72.51 $m^3.s^{-1}$ on 2023/07/22) further contributed to discharge along the Zambezi, with the flow measured at site 44 reaching 212.79 $m^3.s^{-1}$ on 2023/07/24. The highest ADCP discharge was measured at site 29 along the Zambezi River, upstream of Ngoye Falls, with a reading of 2,323.12 $m^3.s^{-1}$ on 2023/05/31. Just two months later, a measurement downstream of the falls at site 45 had decreased to 579.59

$m^3.s^{-1}$ on 2023/07/30. The final discharge measurement at site 47 (567.18 $m^3.s^{-1}$ on 2023/08/06) is comparable to average flows at Katima Mulilo during July (691 $m^3.s^{-1}$) and August (467 $m^3.s^{-1}$).





| Site No. | Site descriptions and river names | Q (m³.s⁻¹) | Date Sampled |
|---|---|---|---|
| 1 | Measured along Lungwebungu River near the source | 8.19 | 2022/06/07 |
| 9 | Measured along Lungwebungu River | 104.91 | 2022/07/04 |
| 11 | Measured along Lungwebungu River | 201.40 | 2023/03/04 |
| 14 | Measured along Lungwebungu River | 288.30 | 2023/03/08 |
| 15 | Measured at end of Kashiji River confluence with Lungwebungu | 75.60 | 2023/03/10 |
| 16 | Measured at end of Lungwebungu confluence with Zambezi | 335.70 | 2023/03/10 |
| 17 | Measured along Zambezi River | 1,471.10 | 2023/03/10 |
| 20 | Measured along Cuando River | 23.91 | 2023/04/06 |





| 21 | Measured along Zambezi River near the source | 1.55 | 2023/05/05 |
| 23 | Measured along Zambezi River, just upstream of Angola-Zambia border | 16.00 | 2023/05/10 |
| 25 | Measured along Zambezi River | 928.56 | 2023/05/17 |
| 26 | Measured at end of Kabompo, confluence with Zambezi | 318.08 | 2023/05/21 |
| 27 | Measured at end of Lungwebungu, confluence with Zambezi | 193.85 | 2023/05/22 |
| 28 | Measured along Zambezi River upstream of Lungwebungu | 1,194.19 | 2023/05/22 |
| 29 | Measured along Zambezi River upstream of Ngoye Falls | 2,323.12 | 2023/05/31 |
| 31 | Measured along Zambezi River, just downstream of Angola-Zambia border | 18.03 | 2023/07/08 |
| 33 | Measured along Zambezi River | 28.52 | 2023/07/13 |
| 35 | Measured at the end of Luisaba River, confluence with Zambezi | 37.72 | 2023/07/16 |
| 36 | Measured along Zambezi River | 105.95 | 2023/07/18 |
| 38 | Measured at end of Chifumage, confluence with Zambezi | 21.17 | 2023/07/21 |
| 39 | Measured along Zambezi River, upstream of Luena | 135.41 | 2023/07/22 |
| 40 | Measured at end of Luena, confluence with Zambezi | 72.51 | 2023/07/22 |
| 44 | Measured along Zambezi River | 212.79 | 2023/07/24 |
| 45 | Measured along Zambezi River | 579.59 | 2023/07/30 |
| 47 | Measured along Zambezi River | 567.18 | 2023/08/06 |

**Figure 5. Selected empirical discharge measurements conducted along the Upper Zambezi segment, with confluence measurements highlighted in grey. See Supporting Material, Table S1, for all ADCP discharge measurements listed in chronological order.**

Despite the limited repeated ADCP measurements and the lack of historical gauge station data within Angola, we estimate the flow contributions from Angola and Zambia at the Lukulu confluence, just upstream of the Barotse Floodplain. Below the Lukulu confluence, several attempts have been made to model flows within the Barotse Floodplain and surrounding areas. However, these efforts are complicated by several factors (see Hughes et al., 2020; Hughes et al., 2023) so we limit our estimates to the upstream section at the Lukulu confluence. The Lukulu confluence represents a mix of flows from the Upper Zambezi, Lungwebungu, and Kabompo sub-basins. Within the Lukulu catchment area, the country split by basin area is: Angola = 57% and Zambia = 43%. ADCP measurements at sites 16 and 17 on 2023/03/10 indicate that the Lungwebungu River contributed 22.82% of the total flow in the Zambezi. However, despite its origins in Angola, the Lungwebungu cannot be considered solely an Angolan river, as a large tributary—the Kasiji River, located within Zambia—contributed 75.60 m$^3$.s$^{-1}$ (measured at site 15), or 22.52%, of the Lungwebungu flow (335.70 m$^3$.s$^{-1}$) on the same day.

ADCP measurements were conducted to estimate the contributions of flows from Angola and Zambia. On 2023/05/10, site 23, located just upstream of the Angolan border with Zambia, recorded a flow of 16.00 m$^3$.s$^{-1}$. By 2023/05/17, at site 24 just below the Chavuma station in Zambia, the flow increased to 923.38 m$^3$.s$^{-1}$. With no significant tributaries between this site and the





Angolan-Zambian border, we estimate that 98.27% of the flow at site 24 originated from Angolan sources. During a separate sampling period, measurements at site 31 (18.03 m$^3$.s$^{-1}$) just downstream of the Angolan-Zambian border on 2023/07/08 and at site 44 (212.79 m$^3$.s$^{-1}$) just upstream of the border on 2023/07/24 suggest a 91.53% contribution from Angolan rivers within the Upper Zambezi segment. At Lukulu, the total estimated flow downstream of the Lungwebungu confluence was 1,388.04 m$^3$.s$^{-1}$ on 2023/05/22, calculated by summing flows from site 27 (193.85 m$^3$.s$^{-1}$, Lungwebungu tributary) and site 28 (1,194.19 m$^3$.s$^{-1}$, upstream of Lungwebungu). The Zambian contributions to this flow include (1) 318.08 m$^3$.s$^{-1}$ from the Kabompo tributary (site 26, with 97.75% of the Kabompo catchment in Zambia). Coincidentally, the Watopa Gauge Station along the Kabompo recorded an average flow of 316.40 m$^3$.s$^{-1}$ for the same date as our ADCP measurement (2023/05/21); (2) 16.00 m$^3$.s$^{-1}$ from site 23 (upstream of the Angolan border), and (3) an estimated 43.66 m$^3$.s$^{-1}$ from the Kasiji River which accounts for 22.52% of the Lungwebungu flow of 193.85 m$^3$.s$^{-1}$, measured previously at site 15. These combined Zambian flows total 377.74 m$^3$.s$^{-1}$, or 27.21% of the flow at Lukulu, with the remaining 72.79% attributed to Angolan contributions during this measurement period.

## 3.3 Water chemistry

A total of nine TWP river expeditions were conducted along the Cuando, Kembo, Lungwebungu, and Zambezi Rivers within the Upper Zambezi segment between May 2018 and August 2023 (Fig. 6a). Although water chemistry measurements were affected by seasonal differences in sampling periods, distinct patterns still emerged from the data. pH levels in the source water regions were generally more acidic, ranging from 4.55 to 5.32, and tended to become more alkaline further downstream (Fig. 6b). This trend was observed along the Lungwebungu River transect in Angola. However, pH levels in the Zambian portion of the Lungwebungu transect were lower, likely due to differences in the sampling season. Despite seasonal variations, the final pH value of the Lungwebungu River before entering the Zambezi was 6.84 (2023/03/10). In contrast, the Kabompo River had a pH of 8.43 (2023/05/21). Along the Zambezi, 10 km upstream of the Kabompo River confluence, the pH was measured at 7.42, increasing to 7.98 below the Kabompo confluence and further to 8.09 below the Lungwebungu confluence, suggesting some mixing but near-neutral conditions during the sampling period. Overall, the rivers tended towards neutral pH levels downstream, with seasonal variation influencing local measurements.






**Figure 6. (a) Water chemistry sampling locations and date of sampling conducted along various rivers in the Upper Zambezi segment; (b) pH measurements; and (c) specific conductivity measurements.**



With respect to specific conductivity, elevated levels were observed at three distinct locations (Fig. 6c). Along the Zambezi, 10 km upstream of the Kabompo River confluence, specific conductivity was measured at 13.78 µS/cm, while the Kabompo

River itself recorded 217.81 µS/cm (2023/05/21), a fifteen-fold increase and the highest specific conductivity concentration measurement across the TWP (Upper Zambezi segment) expeditions. Downstream of the Kabompo confluence, specific conductivity decreased to 89.32 µS/cm and further to 78.27 µS/cm below the Lungwebungu River confluence. The final measurement along the Lungwebungu was 23.60 µS/cm (2023/03/10), nine times lower than the Kabompo levels. Despite differences in sampling periods and seasonal influences, a significant increase in specific conductivity is attributable to the

Kabompo River. This pattern is consistent for both salinity and TDS, with the highest measurements recorded at the Kabompo River Tributary (0.10 PSU and 0.14 ppt, respectively). The Kabompo River tributary exhibited salinity and TDS concentrations sixteen and twenty-one times higher, respectively, than those at the sampling location 10 km upstream along the Zambezi River, and nine and eleven times higher, respectively, than those of the Lungwebungu River.

Elevated specific conductivity was observed during the second Cuando River expedition, conducted at the end of the rainfall season rather than during the dry season. The final measurement in 2018 was 59.99 µS/cm, increasing to 101.47 µS/cm at the start of the 2022 expedition and increased steadily downstream to 163.23 µS/cm. A similar increase was noted downstream of Ngonye (Sioma) Falls, likely reflecting seasonal influences and differences in sampling times. Specific conductivity, salinity, and TDS exhibited similar trends throughout. Both water temperature and DO displayed patterns linked to seasonality. Notable

turbidity perturbations exceeding the WHO limit of 5 NTU were observed in some sections, particularly along the Lungwebungu River. However, turbidity levels decreased to below the 5 NTU limit Lungwebungu-Zambezi confluence. Downstream of Ngonye (Sioma) Falls, turbidity increased sharply, reaching 157.69 NTU near Victoria Falls. See Supporting Material Fig. S3 for additional water chemistry parameters including TDS, salinity, DO, temperature and turbidity.

### 3.4 Wetland and surface water inventory

The CIFOR Tropical and Subtropical Wetlands Distribution V3 dataset from Gumbricht et al. (2017) was used to provide the total extent of each wetland class and percentage cover of wetlands within each HydroSHEDS level 04 sub-basin (Table 6). According to this product, the Upper Zambezi segment contains 52,589 km$^2$ of wetland area in comparison to the Middle (16,133 km$^2$) and Lower Zambezi segments (4,608 km$^2$). The Kameia wetland (described mainly as *Floodplains*) located in Angola, is formed by the Luena, Luateche and Chifumage Rivers of the Upper Zambezi sub-basin, which contains 20,276 km$^2$

of wetlands, covering 21.70% of the total sub-basin area. Notably, other sub-basins originating in Angola including the Lungwebungu (9.87%), Luanginga (18.28%) and Cuando Chobe (5.60%) have extensive wetland coverage, while the Kabompo basin has the lowest wetland coverage (2.34%) within the Upper Zambezi segment. Wetlands cover 9.16% of the Barotse sub-basin and are described mainly as *Floodplains*.





**Table 6. The CIFOR Wetlands inventory from Gumbricht et al. (2017) for each HydroSHEDS level 04 sub-basin within the Zambezi River Basin.**

| Level 04 sub-basin | CIFOR Tropical and Subtropical Wetlands Distribution class areas (km²) | | | | | | | | | | Total Wetland Area (km²) | Total Basin Area (km²) | Wetland cover (%) |
| | Open Water | Man-groves | Swamp | Fens | Riverine and lacu-strine | Flood-outs | Flood-plains | General Marshes | Marshes in arid climate | Marshes in wet meadows | | | |
|---|---|---|---|---|---|---|---|---|---|---|---|---|---|
| **Upper Zambezi Segment** | **481** | **-** | **1,339** | **-** | **5** | **6,926** | **21,734** | **12,589** | **592** | **9,504** | **52,689** | **515,775** | **10.22%** |
| Upper Zambezi | 117 | - | 546 | - | 1.8 | 2,865 | 7,495 | 5,697 | 27 | 3,643 | 20,276 | 93,435 | 21.70% |
| Lungwe-bungu | 24 | - | 187 | - | 0.3 | 315 | 2,220 | 1,131 | 35 | 739 | 4,627 | 46,867 | 9.87% |
| Luanginga | 28 | - | 149 | - | 0.7 | 523 | 4,596 | 1,345 | 65 | 992 | 7,671 | 41,952 | 18.28% |
| Cuando Chobe | 159 | - | 250 | - | 0.1 | 2,897 | 1,737 | 1,646 | 189 | 1,953 | 8,671 | 154,761 | 5.60% |
| Kabompo | 9 | - | 110 | - | 0.5 | 58 | 13 | 1,307 | 71 | 131 | 1,689 | 72,274 | 2.34% |
| Barotse | 144 | - | 97 | - | 1.4 | 268 | 5,673 | 1,462 | 206 | 2,046 | 9,754 | 106,486 | 9.16% |
| **Middle Zambezi Segment** | **5,843** | **-** | **498** | **-** | **10** | **2,582** | **2,239** | **5,946** | **249** | **4,610** | **16,133** | **489,191** | **3.30%** |
| Kariba | 5,089 | - | 157 | - | 6 | 100 | 46 | 211 | 36 | 131 | 688 | 165,211 | 0.42% |
| Kafue | 572 | - | 306 | - | 1.7 | 2,455 | 2,161 | 4,724 | 135 | 4,367 | 14,150 | 155,997 | 9.07% |
| Mupata | 56 | - | 19 | - | 2 | 14 | 21 | 47 | 2 | 24 | 128 | 19,010 | 0.68% |
| Luangwa | 125 | - | 16 | - | 0.3 | 13 | 11 | 965 | 75 | 87 | 1,167 | 148,974 | 0.78% |
| **Lower Zambezi Segment** | **31,828** | **71** | **549** | **4** | **5** | **636** | **51** | **1,945** | **84** | **1,263** | **4,608** | **368,222** | **1.25%** |
| Tete | 2,356 | - | 213 | - | 4 | 289 | 48 | 309 | 32 | 270 | 1,165 | 196,637 | 0.59% |
| Shire | 29,385 | - | 271 | 4 | 1 | 259 | 3 | 1,366 | 34 | 539 | 2,476 | 160,446 | 1.54% |
| Zambezi Delta | 88 | 71 | 65 | - | - | 88 | - | 270 | 19 | 454 | 967 | 11,139 | 8.68% |
| **Zambezi Total** | **38,152** | **71** | **2,385** | **4** | **20** | **10,144** | **24,024** | **20,479** | **925** | **15,376** | **73,429** | **1,373,188** | **5.35%** |

**Table 6 notes. A colour palette is used to emphasise high (blue) and low (red) wetland cover percentages. The spatial distribution of wetlands is mapped in Figure 1.**

While the wetland extent within the Upper Zambezi segment is more than three and ten times larger than that of the Middle and Lower Zambezi segments, respectively, the extent of *Open Water* is much lower. The Middle Zambezi segment contains Kariba Reservoir, Itezhi-Tezhi Reservoir, and Lake Manyeke along the Kafue River, while the Lower Zambezi segment contains Cahora Bassa Reservoir and Lake Malawi (Fig. 2). Of the Level 04 sub-basins in the Middle and Lower Zambezi





segments, only the Kafue and Zambezi Delta sub-basins have significant wetland areas as a proportion of their total sub-basin
area, at 9.07% and 8.68%, respectively.

The JRC global surface water inventory was used to quantify the water occurrence and seasonality of each HydroSHEDS level
04 sub-basin between 1984–2021. Table 7 shows the percentage of pixels indicating water presence in the occurrence band,
while Table 8 presents seasonality data. Both metrics were grouped into ten equal-interval percentage bins, revealing distinct
patterns for individual sub-basins. All sub-basins within the Upper Zambezi segment have >65% surface water presence within
the 1-10% frequency, except for the Kabompo (32.5%). Detailed visual investigation showed several large-scale mine dumps
and pits, constructed in the northern regions of the Kabompo over the past decade, have filled with water, influencing the
surface water dynamics in the sub-basin. The remaining Upper Zambezi segment sub-basins show the highest proportion of
water occurrence within the 1-10% and 11-20% frequency ranges, with the 91-100% frequency signifying permanent water in
rivers and small lakes. The Lungwebungu and Barotse sub-basins have particularly high percentages (5.7% and 5.9%,
respectively) within the 91-100% range.

**Table 7. The JRC occurrence of surface water for each HydroSHEDS level 04 sub-basin within the Zambezi River Basin. A colour palette is used to emphasise high (blue) and low (red) occurrence percentages.**

| Segment | Sub-basin | Occurrence: The frequency with which water was present from 1984 – 2021 (%) | | | | | | | | | |
|---|---|---|---|---|---|---|---|---|---|---|---|
| | | 1-10 | 11-20 | 21-30 | 31-40 | 41-50 | 51-60 | 61-70 | 71-80 | 81-90 | 91-100 |
| Upper Zambezi | Upper Zambezi | 80.4% | 15.0% | 1.5% | 0.5% | 0.2% | 0.2% | 0.2% | 0.2% | 0.3% | 1.6% |
| | Lungwebungu | 81.5% | 5.8% | 1.1% | 0.8% | 0.7% | 0.7% | 0.8% | 1.1% | 1.8% | 5.7% |
| | Kabompo | 32.8% | 12.6% | 10.1% | 8.0% | 9.2% | 8.0% | 5.1% | 2.3% | 3.7% | 8.1% |
| | Cuando Chobe | 70.0% | 22.2% | 4.8% | 0.8% | 0.4% | 0.4% | 0.4% | 0.2% | 0.2% | 0.5% |
| | Luanginga | 87.3% | 10.1% | 0.8% | 0.2% | 0.2% | 0.2% | 0.2% | 0.2% | 0.4% | 0.5% |
| | Barotse | 65.6% | 18.6% | 3.6% | 1.2% | 1.0% | 1.0% | 1.0% | 1.0% | 1.2% | 5.9% |
| Middle Zambezi | Kafue | 59.3% | 8.6% | 2.5% | 1.8% | 1.6% | 1.5% | 1.9% | 2.5% | 4.2% | 16.1% |
| | Kariba | 3.2% | 2.3% | 2.1% | 2.0% | 1.9% | 1.9% | 1.7% | 1.7% | 2.3% | 81.0% |
| | Mupata | 13.4% | 5.8% | 3.9% | 3.3% | 4.0% | 4.4% | 4.7% | 6.0% | 7.8% | 46.7% |
| | Luangwa | 28.3% | 15.4% | 11.3% | 10.0% | 7.8% | 6.3% | 5.2% | 4.3% | 4.9% | 6.5% |
| Lower Zambezi | Tete | 11.5% | 5.7% | 3.7% | 3.0% | 2.8% | 3.2% | 4.1% | 6.3% | 11.4% | 48.3% |
| | Shire | 1.2% | 0.2% | 0.1% | 0.1% | 0.1% | 0.1% | 0.1% | 0.1% | 0.1% | 97.9% |
| | Zambezi Delta | 42.7% | 8.5% | 6.9% | 5.9% | 5.2% | 5.1% | 4.7% | 4.6% | 5.2% | 11.3% |
| | **Zambezi Basin** | **20.3%** | **4.6%** | **1.4%** | **0.9%** | **0.8%** | **0.8%** | **0.8%** | **1.0%** | **1.6%** | **67.7%** |


Seasonality percentages, representing the average number of months water is present in a calendar year (1984–2021), reveal
distinct patterns for individual sub-basins (Table 8). In the Upper Zambezi segment sub-basins, most surface water is present





for 1-4 months of the year, except in the Kabompo Catchment, where 45.9% of inundated pixels are present year-round. In contrast, most surface water in the Middle and Lower Zambezi segments persists year-round.


**Table 8. The JRC surface water occurrence for each HydroSHEDS level 04 sub-basin in the Zambezi River Basin, with a colour palette highlighting high (blue) and low (red) occurrence percentages.**

| | | Seasonality: Number of months water is present within a calendar year, average 1984 – 2021. | | | | | | | | | | |
|---|---|---|---|---|---|---|---|---|---|---|---|---|
| **Segment** | **Sub-basin** | **1** | **2** | **3** | **4** | **5** | **6** | **7** | **8** | **9** | **10** | **11** | **12** |
| Upper Zambezi | Upper Zambezi | 26.5% | 30.9% | 20.9% | 11.4% | 6.3% | 0.8% | 0.3% | 0.2% | 0.2% | 0.1% | 0.0% | 2.4% |
| | Lungwebungu | 30.8% | 29.6% | 17.4% | 7.3% | 2.6% | 0.7% | 0.7% | 0.7% | 0.6% | 0.3% | 0.1% | 9.2% |
| | Kabompo | 9.2% | 4.7% | 4.0% | 3.9% | 4.1% | 5.0% | 5.9% | 6.1% | 8.5% | 2.6% | 0.1% | 45.9% |
| | Cuando Chobe | 43.5% | 21.7% | 14.9% | 9.8% | 3.6% | 1.6% | 0.8% | 0.5% | 0.5% | 0.4% | 0.1% | 2.6% |
| | Luanginga | 35.9% | 35.0% | 19.2% | 5.2% | 1.6% | 0.3% | 0.3% | 0.3% | 0.3% | 0.2% | 0.1% | 1.4% |
| | Barotse | 29.5% | 22.4% | 15.4% | 12.0% | 3.2% | 1.3% | 1.1% | 0.9% | 0.7% | 0.6% | 0.4% | 12.6% |
| Middle Zambezi | Kafue | 20.3% | 21.4% | 10.2% | 3.6% | 2.2% | 1.7% | 1.7% | 2.1% | 2.4% | 1.7% | 0.2% | 32.5% |
| | Kariba | 0.9% | 0.8% | 0.8% | 0.7% | 0.8% | 0.8% | 1.1% | 1.3% | 0.6% | 0.2% | 0.1% | 91.8% |
| | Mupata | 2.5% | 1.9% | 1.6% | 1.6% | 2.0% | 2.5% | 2.1% | 1.8% | 1.5% | 3.6% | 0.3% | 78.6% |
| | Luangwa | 11.8% | 9.4% | 6.4% | 5.3% | 5.6% | 6.1% | 6.4% | 6.2% | 9.5% | 9.2% | 0.6% | 23.7% |
| Lower Zambezi | Tete | 2.2% | 1.9% | 1.6% | 1.3% | 1.2% | 1.2% | 1.9% | 3.2% | 2.9% | 1.3% | 0.4% | 80.9% |
| | Shire | 0.1% | 0.1% | 0.1% | 0.1% | 0.0% | 0.0% | 0.0% | 0.0% | 0.0% | 0.0% | 0.0% | 99.4% |
| | Zambezi Delta | 12.3% | 4.6% | 2.9% | 1.9% | 1.7% | 2.0% | 2.2% | 2.7% | 2.5% | 1.5% | 0.8% | 64.8% |
| | **Zambezi Basin** | **6.4%** | **5.9%** | **3.8%** | **2.1%** | **1.1%** | **0.5%** | **0.5%** | **0.7%** | **0.6%** | **0.4%** | **0.1%** | **78.0%** |

Water occurrence and seasonality were mapped for the Zambezi Basin (Fig. 7), reflecting the distribution of wetlands and open

water across the basin, as shown in Fig. 1. The Kameia, Lungwebungu, Luanginga (Angola); Chobe (Namibia); Barotse, and Kafue wetlands (Zambia); have lower water occurrence and seasonality scores due to annual and seasonal surface water dynamics. In contrast, major lakes and reservoirs show over 90% occurrence and are inundated year-round on average.



**Figure 7. JRC average water (a) occurrence and (b) seasonality from Pekel et al. (2016) for the Zambezi basin from 1984–2021**





The HydroLAKES polygons were clipped to each individual HydroSHEDS level 04 sub-basin (Table 9) within the Zambezi
River Basin. Five surface area intervals were used. The Upper Zambezi segment has the fewest lakes (n = 269), with most (n
= 254) ranging from 0.1 to 1 km$^2$, and Lake Dilolo being the largest (12.14 km$^2$). The Middle Zambezi segment contains over
twice as many lakes (n = 624) as the Upper Zambezi segment. The Kafue (n = 248) and Kariba (n = 304) sub-basins have the
highest number of lakes in this segment. Four lakes exceed 50 km$^2$ in the Middle Zambezi segment: Lake Manyeke (90.29
km²) and Itezhi-Tezhi Reservoir (328.54 km$^2$) in the Kafue sub-basin, Kariba Reservoir (5,276.89 km²) in the Kariba sub-
basin, and Mita Hills Reservoir (58.77 km$^2$) in the Luangwa sub-basin. The Lower Zambezi segment contains 612 lakes, with
most (n = 529) in the Tete sub-basin. Only two lakes are in the Zambezi Delta sub-basin. Four lakes exceed 50 km$^2$: Lake
Manyame (75.18 km$^2$) and Cahora Bassa Reservoir (2,048.68 km$^2$) in the Tete sub-basin, and Lakes Malombe (309.57 km$^2$)
and Malawi (29,544 km$^2$) in the Shire sub-basin.

**Table 9. Number of lakes in each HydroSHEDS level 04 sub-basin within the Zambezi River Basin. See Supporting Material Table**
**S2 for lake counts by country.**

| Basin Segment | No. of lakes/reservoirs in each surface area interval | | | | | |
|---|---|---|---|---|---|---|
| Sub-basin | 0.1 – 1 km² | 1 – 10 km² | 10 – 50 km² | 50 – 100 km² | >100 km² | Total |
| Upper Zambezi | 254 | 14 | 1 | - | - | 269 |
| Upper Zambezi | 83 | 3 | 1 | - | - | 87 |
| Lungwebungu | 17 | - | - | - | - | 17 |
| Kabompo | 7 | - | - | - | - | 7 |
| Cuando Chobe | 16 | 2 | - | - | - | 18 |
| Luanginga | 33 | 2 | - | - | - | 35 |
| Barotse | 98 | 7 | - | - | - | 105 |
| Middle Zambezi | 557 | 59 | 4 | 2 | 2 | 624 |
| Kafue | 212 | 33 | 1 | 1 | 1 | 248 |
| Kariba | 278 | 24 | 1 | - | 1 | 304 |
| Mupata | 24 | - | - | - | - | 24 |
| Luangwa | 43 | 2 | 2 | 1 | - | 48 |
| Lower Zambezi | 565 | 40 | 3 | 1 | 3 | 612 |
| Tete | 497 | 28 | 2 | 1 | 1 | 529 |
| Shire | 66 | 12 | 1 | - | 2 | 81 |
| Zambezi Delta | 2 | - | - | - | - | 2 |
| **Zambezi Basin** | **1,376** | **113** | **8** | **3** | **5** | **1,505** |



## 4 Discussion

### 4.1 Assessment of source regions of the Zambezi River

This study combined ground observation data with several high-resolution, robust EO datasets to quantify and compare the
source regions of the Zambezi River. The use of EO datasets have enabled geospatial assessments of this vast area (Zambezi basin: 1,370,000 km$^2$) which lacks evenly distributed, long-term hydrological and meteorological monitoring data across the basin. Our combined estimates of river length, flow accumulation, precipitation, historical and empirical discharge, seasonal flows, water chemistry, wetland inventories and surface water dynamics reveal five significant contributions from rivers originating within Angola that have been overlooked:

1. The Lungwebungu River is the longest tributary of the Zambezi River according to empirical expedition track data.

2. The Angolan portion of the Upper Zambezi sub-basin is the highest contributor to river discharge according to historical gauge station records and empirical ADCP measurements.

3. The Lungwebungu and Upper Zambezi sub-basins of Angola are shown to improve water quality, reducing elevated specific conductivity, salinity and TDS levels—attributed by inputs from the Kabompo sub-basin of Zambia.

4. The Upper Zambezi sub-basin has the highest proportion of source wetlands, originating from rivers and drainage lines within Angola.

5. This empirical confirmation highlights the pioneering assessments of White (1983), Jackson (1986) and others regarding the significance, including the biodiversity and ecosystem services, of the Zambezian Grasslands biome, centred on the reservoir rivers of the Upper Zambezi Floodplains Ecoregion (Skelton 1994, 2024).


Complementary definitions of the river source concept use river length together with characteristics of individual tributaries, which represent respective hydrological attributes of the river system. In this respect, we provide evidence that the currently recognised source river of the Zambezi River is not the longest tributary, ranking 5th. Empirical comparisons of river length track data and high-resolution digitisation reveals the Lungwebungu River is approximately 342 km longer than the
traditionally recognised source. Originating in Angola, this river is the source of the Zambezi according to the criterion of river distance (Fig. 2). In comparison, although the 1,387 km Manyame River in Zimbabwe starts at the highest elevation (1,660 masl), it originates from a separate headwater region, distinct from the most distal source water regions: the Lufilian Arc and the AHWT. Historical gauge station data indicates that each of the major tributaries in the Upper Zambezi Segment is perennial, and therefore, no alternative ephemeral water sources were identified within our analysis at this scale.


ADCP measurements conducted along the Zambezi River indicate that the Angolan contribution to the Upper Zambezi sub-basin (Fig. 2) was 98.27% in May 2023 and 91.53% in July 2023. The measurements show that in July 2023, the Luena River tributary was the highest contributor, providing approximately 34% of the discharge during that period. ADCP measurements were conducted to estimate the flow contributions from Angola and Zambia at the Lukulu confluence, located downstream of



the Upper Zambezi, Lungwebungu and Kabombo sub-basins, and upstream of the Barotse Floodplain (Fig. 2). Estimates show that in May 2023, 27.21% of the flow at Lukulu could be attributed to Zambia, with 72.79% attributed to Angola. The Lukulu catchment, comprising Upper Zambezi, Lungwebungu and Kabombo sub-basins, is divided by area: Angola = 57% and Zambia = 43%. Although the Angolan portion covers only 14% more area, it contributes approximately 45.58% more flow at the Lukulu confluence than the Zambian portion. Developments or impacts on the Angolan portion of the Lukulu catchment

will therefore disproportionately affect the rest of the Zambezi River system. Further flow measurements over extended periods is necessary to account for seasonal variations.

The water chemistry parameters (water temperature, specific conductivity, TDS, salinity, pH and DO) of the Cuando, Kembo and Lungwebungu and Zambezi Rivers fall within WHO (2011) potable water limits. This highlights the natural high water

quality conditions of these source rivers, with minimal impacts on their upstream water quality (Huntley et al., 2019). However, the Kabombo River has elevated specific conductivity, salinity and TDS levels. Inflows from the Upper Zambezi and Lungwebungu sub-basins reduce these elevated levels, with the Kabombo showing levels more than nine times higher than those in the upstream Zambezi and Lungwebungu Rivers for each of these three parameters. The Kabombo catchment in Zambia hosts several mining operations, particularly for copper, cobalt, and zinc (Steven, 2000; Mukube and Syampungani,

2025). These activities are expected to increase salinity, TDS, and conductivity in nearby water sources due to the release of contaminants such as sulphates and heavy metals (WHO, 2011). This can increase dissolved substance concentrations, impacting regional water quality, however, further sampling during different seasons is still required.

The Lungwebungu River begins as a small trickle from a circular ombrotrophic peatland bog (See Supporting Material Fig.

S4), while the traditional source of the Zambezi originates from a forest spring (See Supporting Material Fig. S5). Supported by wetlands and groundwater, these sources flow year-round. The Upper Zambezi sub-basin has the highest wetland coverage (21.70% or 20,276 km$^2$), with 94.61% (19,184 km$^2$) in Angola, sustaining overall river health. Notably, we highlight the little-documented Kameia wetlands and Luena Flats within Moxico Province, primarily described as *Floodplains* (See Supporting Material Fig. S6b). The Luena Flats, adjacent to the Luena River, are crucial for local fishermen and feature a distributary

channel system where seasonal flooding drives river meander migration, typical of rivers in this region (Mendelsohn, 2022). Spanning approximately 25,000 km$^2$, these wetlands are shaped by seasonal river discharge and overbank flooding.

## 4.2 Hydrology

The Zambezi River Basin, the largest in southern Africa, has a cumulative mean annual flow of approximately 97 km$^3$ from

its main river and tributaries, broadly comparable to that of the Nile (Moore et al., 2022). The hydrology of the Zambezi Basin is shaped by seasonal rainfall patterns that are consistent over long timescales, though water availability and evaporation vary with latitude (Beilfuss, 2012). In the northern water tower and headwater regions, precipitation is higher (~2000 mm/year) and





more consistent, while in the southern sections, it is lower (~450 mm/year). Northern tributaries contribute the most to runoff (Meier et al., 2011). Water withdrawals for irrigation, industrial, and domestic use are estimated to constitute only 3% of the

Zambezi's total flow, while evaporation from major reservoirs accounts for slightly more than 5% (Moore et al., 2022). The literature highlights the anticipated overall water scarcity and its potential impact on future hydropower generation (Arias et al., 2022; Dube and Nhamo, 2023).

McCarthy et al. (2000) state that the Zambezi's annual flows follow an 80-year cycle of alternating wet and dry periods, a

long-term pattern also observed in the Okavango River. Historical records of flooding in the Okavango Delta show average to above-average outflows from 1849 to 1900, followed by reduced outflows with some favourable years from 1900 to 1951. During the 1960s and 1970s, outflows were above normal, whereas the 1980s and early 1990s experienced below-normal outflows (McCarthy et al., 2000). Records of water level from the hydrological station at Victoria Falls, collected since 1907 indicate a similar pattern: below normal from 1907–1946, above normal from 1947–1981, and again below normal from 1982–

1997 (Moore et al., 2022). The implication and significance is that the Okavango and Zambezi Basins share the same headwater region: the AHWT (Lourenco and Woodborne, 2023). This evidence shows that fluctuations in precipitation in the shared headwater (water tower) region of the Okavango and Zambezi Basins have significant and widespread effects across both river systems.

Several investigations into modelling the hydrological flow regimes and discharge of the Zambezi River Basin have been conducted (Meier et al., 2011; Zimba et al., 2018; Hughes et al., 2020; Hughes et al., 2023). Despite the inherent uncertainties in river discharge modelling, the Upper Zambezi segment presents significant challenges due to its extensive wetlands, emplaced across deep Kalahari sediments, for which an understanding of the water exchange dynamics between river channels and wetland storage areas is basically non-existent (Hughes et al., 2020). A comparison of long-term gauge station data with

datasets from Collins et al. (2024) and Akpoti et al. (2024) highlights these uncertainties. Hydrological modelling remains particularly difficult in data-scarce conditions (Meier et al., 2011), and the absence of hydrological monitoring stations on Angolan rivers further exacerbates data scarcity in the region. Historical discharge data from the Zambian sections of the Upper Zambezi segment indicate that the Upper Zambezi sub-basin contributes the highest flow, particularly during the wet season, measured at Chavuma (mean annual discharge 637 $m^3.s^{-1}$). The Lungwebungu and Kabompo sub-basins provide further inputs,

particularly during the dry season, measured at Lukulu (mean annual discharge: 745 $m^3.s^{-1}$: Moore et al., 2022). The contributions of the Barotse and Luanginga provide increased flows measured at Sananga (mean annual discharge: 1,040 $m^3.s^{-1}$: Hughes et al., 2020), whereas the Cuando Chobe basin has a smaller influence (mean annual discharge: 33 $m^3.s^{-1}$ Schleiss et al., 2017; Pallett et al., 2024). Mean annual discharge at the Victoria Falls hydrological station increases to 1,171 $m^3.s^{-1}$, indicating that most of the flow originates upstream of the Barotse Floodplain, with little additional water reaching

Victoria Falls after the Barotse and Luanginga sub-basins.



The Cuando Chobe River sub-basin, despite having the highest flow accumulation pixel count, has a limited impact on the Zambezi's average yearly runoff (Schleiss et al., 2017). Its size is the reason for over-estimations of modelled discharge, but under present climatic conditions, it does not contribute to Zambezi hydrology significantly (Schleiss et al., 2017; Pallett et al., 2022). The flow of the Cuando River itself has been measured at Kongola (17.8123° S, 23.3846° E), Namibia since 1969 (Pallett et al., 2022). The flow is relatively constant, having significant peaks during times of intensive rainfall (Pallet et al., 2022). The Collins et al. (2024) discharge data revealed an average peak of 1,845 $m^3.s^{-1}$ in February and lows of 40 $m^3.s^{-1}$ in September, while discharge measured at the Kongola gauge station indicates peaks of 39 $m^3.s^{-1}$ in July and lows of 27 $m^3.s^{-1}$ in December (Pallett et al., 2022), indicating that the discharge values from the Collins et al. (2024) dataset for the Cuando River are (1) inflated and (2) peak flows occur out of season. This discrepancy reflects the relatively slower water passage through the Cuando River's vast 5 to 15 km-wide floodplains covering approximately 3,450 $km^2$ over a distance of just over 500 km (Pallett et al., 2022). The marshy floodplain is covered in tall grasses, phragmites reeds and papyrus which results in evaporative losses, seepage and delays in flow. Consequently, the Collins et al. (2024) and Akpoti et al. (2024) datasets should be used cautiously in this region, as they both fail to account for evaporative processes, infiltration, and residence time. In addition, the elevated specific conductivity, salinity and TDS measured along the Cuando can be attributed to the high rates of evaporation.

### 4.3 Implications for transboundary river management

Since 2004, ZAMCOM has worked towards the equitable and reasonable utilisation of water resources through efficient management and sustainable development. However, due to its remote inaccessibility, little attention has been paid to the AHWT. Angola is often referred to as the "water tower" of southern and south-central Africa (Huntley, 2019), yet the geopolitical importance of the AHWT remains understated in discussions about the future of the Zambezi River. Apart from a few ephemeral rivers and local rainfall, the Zambezi receives no significant inputs between the Barotse floodplain and Victoria Falls (with the slight exception of the Cuando and Luanginga rivers, both of which originate in AHWT). We can broadly state that approximately 70% of the water entering the Kariba Reservoir, below Victoria Falls, originates in the AHWT, with pivotal—yet previously overlooked—implications for transboundary river management. This evidence extends to transboundary systems supported by the AHWT, an estimated 95% of the Okavango Delta's inflow originates from precipitation in Angola (Folwell et al., 2006). The Upper Zambezi and Okavango basins constitute the primary dry-season water supply for the Kavango-Zambezi Transfrontier Conservation Area (KAZA-TFCA), the world's largest TFCA, which protects keystone regional wildlife populations (KAZA-TFCA, 2019). Pioneering assessments by White (1983), Jackson (1986), and others highlight the significance of the Zambezian Grasslands biome and tributaries of the Upper Zambezi as some of Africa's reservoir rivers. The Upper Zambezi tributaries typically exhibit minimal fluctuation in water levels due to the slow release of water, making them a vital hydrological resource (Jackson, 1986).



Water towers refer to the water supply and storage provided by mountain ranges to support and sustain environmental and human water demands downstream (Immerzeel et al., 2020). They are recognised for their buffering capacity, where water—typically stored in glaciers, lakes, and groundwater—provides a steady supply to downstream areas (Immerzeel et al., 2020). Unlike cryospheric water towers, the AHWT (and other African water towers) are sustained by unique buffering mechanisms, in the absence of permanent snow and ice (Lourenco and Woodborne, 2023). Underlain by a large Kalahari sediment aquifer

and flanked by extensive miombo woodlands, groundwater-fed peatlands grow within the dambo environment of the Zambezian Grasslands biome. These natural sponges absorb, filter, and store water during the wet season, and gradually release it into the river system during the dry season (Lourenco et al., 2022). Radiocarbon dating of active riparian peatlands suggests peatland initiation around 7,100 cal. yr BP (Lourenco et al., 2022), after the onset of the African Humid Period during the early Holocene (Shanahan et al., 2015)—a testament to the long-term climate resilience of the AHWT. The El Niño Southern

Oscillation (ENSO), arguably the world's most significant interannual climate phenomenon (Timmermann et al., 2018; Cai et al., 2021) has important ecological and socio-economic impacts in the Okavango and Zambezi Basins, making the long-term buffering effect of the AHWT crucial to the long-term resilience of people, wildlife, and ecosystems downstream. However, the AHWT remains sensitive to disturbance, particularly from mining and deforestation. Erosion caused by these activities would destabilize the natural buffering mechanisms and hydrology of the AHWT.


Future development in the Zambezi River Basin is inevitable, but it must be carefully managed to balance the conflicting demands of humans and environmental conservation. Our findings highlight the stark contrast between surface water dynamics in impounded versus free-flowing rivers. In the Upper Zambezi, floodplains with dense vegetation and gentle slopes act as natural sediment traps, limiting downstream transport and supporting vital wetlands for fish, birds, and mammals (Schleiss et

al., 2017; Moore et al., 2022). These tributaries also provide a reliable perennial water source, sustaining human settlement and agriculture (Pallett et al., 2024). The construction of several major reservoirs downstream of Victoria Falls, including Kariba, Cahora Bassa, Itezhi-Tezhi, and Kafue Gorge, has led to significant changes in the ecology, economic significance, and use of the Zambezi River system (Moore et al., 2022). Given ongoing development, future water demand is set to increase, with consequent pressure on authorities in the region to meet it. How this increasing demand is met will be critical for the

Zambezi basin: development projects in key headwater regions, such as Lungwebungu and the Upper Zambezi sub-basins, will destabilise the region's water security. Angola is developing rapidly, with expanding road and rail infrastructure providing access to the AHWT. Given its relatively low population density (Lourenco and Woodborne, 2023), special attention must be paid to sustainable development that maintains current water flows and chemistry in the Upper Zambezi. Water availability and contamination are transboundary challenges that require high-priority, coordinated action to safeguard the basin's future.


The construction of dams has become crucial for regional hydroelectric power, while large reservoirs have boosted tourism and contributed to local economies (Khan et al., 2014; Arias et al., 2022). However, economic benefits have come at significant human and ecological costs, the construction of Kariba and Cahora Bassa reservoirs displaced thousands of people and caused



lasting community and psychological impacts (Moore et al., 2022). Ecologically, the reservoirs have disrupted major
floodplains by reducing water and sediment flow, leading to the contraction of wetlands and adverse effects on biodiversity
(Moore et al., 2022). Sediment reduction from Cahora Bassa has led to erosion and loss of soil fertility in the Zambezi Delta,
impacting agriculture and wildlife (Ronco et al., 2010; Khan et al., 2014). Degradation of the Zambezi Delta illustrates how a
reservoir upstream of the Barotse Floodplain would negatively impact a system dependent on a seasonal flood pulse (Hughes
et al., 2020; Hughes et al., 2023). Reduced seasonal water flow and sediment supply will alter wetland processes, decreasing
wetland extent, channel width, and drought resilience. Increased reservoir evaporation will raise salinity, promote harmful
algal blooms, and disrupt water chemistry, affecting pH and oxygen levels. Furthermore, migration pathways of aquatic species
will be broken, with potential basin-wide consequences, as headwater and wetland regions serve as a biological reserve,
providing breeding and recruitment sites. A secondary consequence will be increased wetland loss as drying vegetation
becomes more prone to fires and conversion to agriculture.


Climate change is expected to reduce water availability in the Zambezi Basin, which, already prone to extreme floods and
droughts, will face even more challenging conditions due to its distinct wet/dry seasons and reliance on hydropower (Fant et
al., 2015; Schlosser and Strzepek, 2015; Arias et al., 2022). This is particularly critical in Africa, where increasing climate
extremes threaten limited water infrastructure (Cervigni et al., 2015). The current natural water chemistry and flows of the
Angolan tributaries enhance the Zambezi Basin's resilience to climate shocks and provides critical ecosystem services to the
basin at large. The Angolan tributaries are described as some of Africa's least affected by socio-economic development and
remain relatively underdeveloped due to the historical impact of war, which slowed social and economic progress (OKACOM,
2023). The conservation of this area will ensure these ecosystem services continue; however, given the realities of growing
demand for economic development in the region, Angola cannot continue to bear the full economic cost for the basin. Greater
recognition of these Angolan source waters is needed—they are ensuring regional water security. The governments of the
Zambezi Basin must collaborate across departments and sectors to holistically manage the basin's waters and ensure the
region's water future.

## 5 Conclusion

By combining in-field data with EO datasets, this study reveals that the Angolan contribution of southern Africa's largest river
has been overlooked. The Angolan Lungwebungu River is the Zambezi's longest tributary. It is significantly longer (342 km)
than the currently recognised source near Mwinilunga, Zambia. The Lungwebungu and Upper Zambezi sub-basins of Angola
are shown to improve water quality, reducing elevated specific conductivity, salinity and TDS levels attributed by inputs from
the Kabompo sub-basin of Zambia. The Upper Zambezi sub-basin, particularly its Angolan component, contributes the most
to river discharge and contains the highest proportion of wetlands. These wetlands, including the Kameia wetlands and Luena
flats, support significant ecological functions and local fisheries. In their combination, greater river length, higher discharge,



water quality and wetland coverage position the Angolan rivers as the more significant source of the Zambezi River. This study highlights the need for improved monitoring and analysis of flows across the Upper Zambezi, particularly the Angolan tributaries. In addition, Angola deserves acknowledgment for its efforts in hosting the source and main aquifer of the Zambezi River. The natural flow patterns facilitate essential ecological processes which are vital for sustaining aquatic and terrestrial

life. Any developments, including dam construction, wetland transformation, or water diversion projects, must carefully assess their potential impact on natural flow patterns and ecosystem services.

**Acknowledgments**

The Wilderness Project, National Geographic Okavango Wilderness Project, and Wild Bird Trust expedition teams are acknowledged for their sample collection conducted during the Zambezi River expeditions. David Garret and Dr. Joe Cutler

are acknowledged for their discussions and feedback throughout the writing process. The Zambian Water Resources Management Authority is acknowledged for sharing the historical gauge station data used in this research.

**Code and data availability**

The code and data supporting the findings of this study are available upon reasonable request from the corresponding author. Please contact Mauro Lourenco (mauro@wildbirdtrust) for further details.

**Author contribution**

**RvB** and **RSB** conceptualised the study and designed the research methodology. **ML**, **FPDC** and **TF** conducted the formal analysis. **ML** produced all the figures and tables. **RvB** and **RSB** carried out the investigation and fieldwork. **ML, FPDC, TF, FCN and GJR** contributed to the data validation and data curation. **ML** was responsible for writing the original draft. **RvB** supervised the research. All authors reviewed, edited and approved the final version of the manuscript.



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
