# Peer review of "Assessment of source regions of the Zambezi River: implications for regional water security"

_EGUsphere, 2025_

## Author Comment (AC1)

**Author's response to Reviewer 1 comments**

EGUSPHERE-2025-837 | Research article

**Manuscript Title: Assessment of source regions of the Zambezi River: implications for regional water security.**

The authors would like to thank the anonymous reviewer and the editor (Professor Hubert H.G. Savenije) for their time and effort in providing their expert views and comments on the manuscript during the review process.

As per the journal's instructions, we are currently in the final response phase and are providing detailed replies to all referee and community comments. A revised manuscript has not yet been submitted, as this is not requested at this stage of the review process. Where relevant, we describe changes to the manuscript by referring to specific sections or content rather than line numbers or tracked edits, since the revised manuscript is not provided at this stage.

We have made every effort to address the points raised, as detailed below.

**Reviewer #1: Citation:** https://doi.org/10.5194/egusphere-2025-837-RC1

| Overall comments | Authors response |
|---|---|
| Lourenco et al. through this contribution wish to clarify the hydrology, ecology and climatology of the Zambezi, as one of the most important basins in Southern Africa. Understanding these processes including their interactions is crucial, especially now under the current rapidly developing conditions in the basin. There is a large interest in mining, further use of water for food, and hydropower, all likely impacting in different ways on the Zambezi's water resources. As seen from last year's drought, exclaimed (in Zambia) to be the largest in 20 years time, climate variability will pose a large challenge as well. In order to protect, sustain and distribute water resources in an equitable manner, any developments should be underpinned by scientific data and analysis, that predicts the likely impact of the development in conjunction with climate change and climate variability.

Now did Lourenco et al. succeed in making a contribution to this end? I have quite struggled to get through the paper and to find the most prominent merits. The | We thank the reviewer for their thoughtful summary of the importance of understanding the Zambezi Basin's hydrology, ecology, and climatology in the context of development, climate change, and transboundary water management. We agree with the reviewer's framing of the basin's significance and the urgent need for robust, data-driven insights to inform decision-making.

We respectfully contest the assertion that the manuscript lacks scientific contribution or novelty. Our study brought together the best available empirical, remote sensing, and modelled data to assess hydrological processes across the Upper Zambezi and its tributaries (particularly in Angola), where data is extremely limited. To our knowledge, this is the first attempt to:

1. Empirically traverse and map the Zambezi River from both its Zambian and Angolan sources, using GPS-tracked ground expeditions. These transects were used to quantify and compare river lengths and inform the hydrological and geomorphological understanding of the headwaters.
2. We present new ADCP-based flow measurements across several key tributaries, particularly in Angola, where no prior discharge data has been published or reported. These are the only known field-based discharge data for many of these sites. This flow data was used to quantify the Angolan contribution to the Zambezi by combining them with historical gauge station data.
3. We evaluated two of the most recent EO-based modelled discharge products (Collins et al., 2024, and Akpoti et al., 2024) across the Upper Zambezi sub-basins, and identified limitations and discrepancies when compared to our ADCP data.
4. Provide a detailed inventory of wetlands and the seasonality and occurrence of surface water, highlighting the natural (undammed) hydrological behaviour of the Upper Zambezi, particularly within Angola, which has received limited attention in the literature compared to more studied wetland regions in Zambia. |

| | |
|---|---|
| paper's English is likely better than my own, but for a scientific contribution, there is too much data, there is also too much data that is not new and also not analyzed in a novel way, and many data also do not lead to new evidence or quantifications of hydrological processes or other key hydrological knowledge, which would make it a logical contribution for this journal. The paper reads as a report rather than a scientific article. | We have revised the manuscript for clarity, focus, and scientific framing. We have shortened sections to distinguish between methods, results, and discussion more clearly. We emphasise on the research questions and novelty in the introduction and conclusion. We also provide a clearer explanation of how multiple data sources were integrated to validate our findings. |
| | We acknowledge that this work is foundational rather than definitive, serving as a call for further research into the *whole* Upper Zambezi and, crucially, for long-term monitoring. We advocate for the establishment of gauging stations, particularly in Angola, and the maintenance of existing stations in Zambia, to build upon this first step in quantifying the hydrology of this critical region. |
| | We hope the responses, revisions and clarifications help demonstrate the scientific value of our contribution and its relevance to basin-wide hydrological understanding and sustainable management. |
| I start here with commenting on the 5 key findings from the discussion:

l. 613-614 "...reveal five significant contributions from rivers originating within Angola that have been overlooked". This already sounds as if nobody ever thought of the fact that a significant portion of water from the Upper Zambezi comes from Angola. The well known and widely used HydroSHEDS dataset already shows that a very large portion of the upstream area comes from this area. This is quite trivial and none of the data collected and shown gives very new insights. On the 5 key points below that: | We acknowledge that the role of Angola in contributing to Upper Zambezi flows is not unknown, and datasets such as HydroSHEDS have indeed long suggested this. Our intention was not to imply that this hydrological contribution has never been recognised, but rather that its importance, scale, timing, and implications have been underexplored in a quantified, integrated, and cross-disciplinary manner.

While geographic datasets clearly delineate the catchment boundaries, our study represents the first attempt to physically traverse, empirically assess, and synthesise hydrological data from both the Zambian and Angolan sources of the Zambezi.

This includes: Field-based quantification of discharge using ADCP measurements for several Angolan tributaries, many of which had no previous in-situ flow data; Assessment of river length and morphology, providing empirical evidence of the Lungwebungu as the longest headwater source of the Zambezi; Integration of water quality, surface water occurrence and seasonality, and wetland extent data to provide a more comprehensive assessment of the ecological and hydrological significance of the Angolan headwaters; Comparison with modelled datasets (Collins et al., 2024, and Akpoti et al., 2024), revealing important discrepancies and reinforcing the need for field validation.

Thus, while the contribution of Angolan tributaries may be broadly recognised on maps, this study is the first to attempt an integrated, data-driven evaluation of their actual hydrological and ecological importance. We have revised the language in the manuscript to clarify that we are not claiming the geographic origin itself is a novel finding, but rather that our empirical investigation of these headwaters and their characteristics is a significant and previously underappreciated step toward understanding the Zambezi basin. |
| 1. The length of the Lungwebungu tributary: the Lungwebungu may be longer than what is currently considered as main stem, but this length has long been known from terrain analyses. I do not see any new insights here. There may be other reasons (as the | We appreciate the reviewer's point regarding the availability of terrain analyses that could be used to reveal the Lungwebungu's length. However, we respectfully disagree that our findings offer no new insights or are merely narrative in nature.

To our knowledge, no published article or scientific report has explicitly quantified the Lungwebungu's length. While terrain data might allow for |

| | |
|---|---|
| authors show in their own literature review) to call the origin of a certain stem the "source", from a hydrological or water resources point of view, this is not relevant. It is only a narrative. | this quantification, existing models often misrepresent river length due to issues of scale, resolution, and meander underestimation, which we demonstrate through our own river length comparison analyses (Table 3).

Our work represents the first known full-length, manned scientific transect of both the Lungwebungu and the Zambian headwater source, extending all the way to the Indian Ocean. These expeditions provided direct, ground-truthed evidence on river morphology, connectivity, flow characteristics, and navigability, critical factors often assumed in remote basins but rarely verified.

We acknowledge that defining a river's source can have geographical, hydrological, and even political or cultural dimensions (as shown in Table 1). However, in fluvial geomorphology and hydrology, it is widely accepted that the source corresponds to the longest continuous tributary. This is a standard, objective criterion, not a narrative one. By this definition, our findings clearly confirm the Lungwebungu as the Zambezi's source based on the river length criterion, which has important implications for basin delineation, hydrological modelling, and upstream water contribution assessments.

We have revised the manuscript to ensure the framing reflects this contribution more clearly and avoids overstating the novelty where it is not warranted, while still highlighting the substantial empirical value of this data. |
| 2. The quantification of the Angolan contribution. Probably it is the largest, but as said, is that really a novel finding? Also, the empirical gauge records used to support this, cannot provide any evidence whatsoever, as observations were only snapshots, and even not taken at the same moment. I understand that taking measurements at the same moment is not possible, (and appreciate how difficult it is to collect these) but this "new insight" is thus not supported by data. | We fully acknowledge that the role of Angola as a major source region for Zambezi headwaters is not entirely unknown. However, to our knowledge, no prior study has attempted to empirically quantify this contribution using a combination of in-situ measurements, historical gauge station data and modelled river discharge estimates. While datasets such as HydroSHEDS illustrate the spatial extent of upstream areas in Angola, they do not provide direct flow estimates specific to these sub-catchments. Our results help close that gap.

Our aim was not to claim the existence of Angolan contributions as a new discovery, but rather to provide the first integrated, empirical attempt to characterise their magnitude and importance; this was done within the limitations of available data.

We agree with the reviewer that snapshots alone cannot fully support definitive flow contribution estimates across seasons or years. However, we have clearly stated in the manuscript that these ADCP-based observations represent an initial baseline and should not be interpreted as long-term averages. Moreover, the measurements were combined with historical gauge station data where available. Adding further weight to our interpretation.

Importantly, our manuscript does not claim that this analysis alone settles the hydrological dominance of the Angolan headwaters. Instead, we explicitly state that these findings highlight the need for long-term monitoring infrastructure (particularly in Angola) and urge future research to validate and build upon this initial assessment. We have revised the manuscript to ensure that this clarification is accurately reflected. |
| 3. Provided only data taken at the same moment (2023/05/21 for the Kabompo), and the fact that a 15 times higher concentration was found in the Kabompo, this finding is to my | We appreciate this comment and are satisfied that our data provides a direct and temporally aligned comparison, strengthening the reliability of the observed 15-fold increase in conductivity, above and below the Kabompo tributary. |

| | |
|---|---|
| mind supported sufficiently by data. The Kabompo lies in the middle of the Copperbelt region, and is subject to a lot of mining activities, which may be the cause of this high salinity and other contaminants. The Lungwebungu concentration however, was taken during the rainy season, and this measurement is therefore difficult to compare. During this season, flows are likely (much) higher than the dry season, hence this can dilute contamination sources, rendering any conclusions on this matter false. | We also fully agree that seasonal variation can significantly influence water quality through dilution and hydrological flushing. As the reviewer notes, the last Lungwebungu water chemistry reading sample was taken during the rainy season, while the Kabompo measurement occurred later, during lower-flow conditions. We note the dilution from the Lungwebungu in our results, with specific conductivity decreasing further to 78.27 µS/cm in the Zambezi river, below the Lungwebungu River confluence.

In addition, the criticism is incorrect. The Lungwebungu was sampled in the dry season as that river expedition was carried over two separate dates, Angola: June-July 2022 and Zambia: March 2023. Mean conductivity measured within the Angolan section (during the dry season) was 12.28 µS/cm (n=77) SD 1.15 with a maximum value of 14.95 µS/cm. These values are significantly lower than those recorded in the Kabompo in May 2023 (218 µS/cm). They reinforce the observation that the Kabompo displays anomalously high conductivity. Moreover, June and July fall in the middle of the dry season, a time when conductivity is expected to be at its highest due to reduced dilution and increased evaporative concentration. The Kabompo measurement in May, marks the start of the dry season, and thus, the measured conductivity of 218 µS/cm is likely below its seasonal peak.

We would like to clarify, however, that our intent was not to present this comparison as a definitive, seasonally controlled analysis, but rather as an initial observational contrast that raises important questions about localised impacts of land use, particularly mining, on water quality in Zambia. Additionally, The Wilderness Project is currently conducting (having started on 3 June 2025) another river expedition specifically along the Kabompo River in Zambia, with the intention of reporting further details regarding potential water chemistry concerns on this major tributary.

We have revised the discussion to further highlight the preliminary nature of the comparison. We also emphasise that this finding aligns with broader concerns about anthropogenic pressures in the Kabompo/Copperbelt region, which are well-documented. Additionally, we reiterate in the conclusion that longitudinal and multi-seasonal monitoring is essential, and advocate for this as a next research step rather than claiming conclusive outcomes from these initial data. |
| 4. The proportion of wetlands. The authors only reduced the wetland map over subbasins, this is hardly a new finding. It is only a summary of already existing data sources. | We appreciate the reviewer's point and have stated in the manuscript that the wetland maps themselves are derived from existing sources. However, we respectfully argue that our analysis provides a novel synthesis and hydrologically relevant interpretation of these data in a way that has not previously been done.

Specifically: We quantify the proportion of wetland area within each sub-basin of the Upper Zambezi for the first time, using a consistent spatial framework aligned with contributing tributaries. This allows us to link wetland distribution to upstream water sources, identifying which tributaries sustain the region's most extensive wetlands and floodplains. Prior maps and datasets do not provide this basin-structured perspective, nor do they assess the implications of wetland extent for hydrological buffering, and flow dynamics, specifically for the Angolan tributaries.

This level of sub-basin wetland attribution is essential for integrated water management, particularly under increasing development and climatic pressures in the Zambezi system. We have revised the text to make this |

| | analytical contribution clearer and to more explicitly distinguish it from existing mapping efforts. |
|---|---|
| 5. The highlight of significance: this is not really a finding, merely a confirmation what was already known. | We appreciate the reviewer's perspective. However, we respectfully argue that while certain aspects of the Angolan contribution to the Zambezi system may have been recognised qualitatively or anecdotally in past literature and datasets (e.g., HydroSHEDS), they have not been systematically quantified or integrated across multiple data sources, including terrestrial surface water storage, wetland extent, flow measurements, and water quality, as we do here.

Our intention was not to claim the existence of Angolan Highlands Water Tower contributions as a new discovery, but rather to present our new data to demonstrate its scale, multidimensional character, and under-recognised importance in shaping downstream hydrology and ecology. |
| So all in all, despite the length of the paper, I find it not strong and not novel. To my mind, the strongest contribution lies in the water quality sampling. This was done in a season with relatively low flows, and with the order of magnitude difference in concentration measured, I think the conclusion that the Kabompo is negatively impacting the water quality of the Zambezi is robust. If this paper is to be accepted, I would recommend focusing on this finding, only provide detailed description of data and analysis that supports findings (you may mention which data you considered and then quickly conclude why they were not suitable), and deepening the water quality aspect out with further measurements that can also give insights in the variability of the quality in time. This may the lead to proper quantification and possibly scientific support for policy on further developments or conservation in the Kabompo basin, and/or development and conservation of the Angolan part as a contributor to improved water quality. The collaboration with the Zambia Water Resources Management Authority is crucial for this. | We thank the reviewer for their constructive suggestion and for recognising the strength of our water quality analysis. We agree that the marked contrast in conductivity and potential contamination between the Kabompo and Angolan tributaries is a valuable and policy-relevant finding.

However, we respectfully disagree with the recommendation to refocus the manuscript exclusively around this point. Our central aim is to present a multi-dimensional reassessment of the Upper Zambezi headwaters, integrating:

Hydrological contributions from Angola, which remain underrepresented in basin-scale narratives, river length quantification (explained above in connection with geomorphological definitions of a river's source), wetland mapping, which connects source regions to flood-dependent ecosystems, and initial, but telling, water chemistry contrasts, which we see as one emerging dimension of broader upstream-downstream differences.

These elements are not isolated observations, but rather complementary lines of evidence that together support a more integrated, transboundary understanding of how the headwaters shape water quantity, quality, and ecosystem resilience further downstream.

We acknowledge, however, that the manuscript is dense in detail, and we have responded by shortening sections that might be construed as overly long. Additionally, we have revised the text to focus more directly on our findings and their relevance to the literature, rather than broader statements or general observations about the Zambezi Basin as a whole.

We also emphasise that this study represents an important first step. We fully agree with the reviewer's suggestion that this work be extended through longitudinal sampling campaigns and the need for collaboration. We agree that collaboration is essential, but we emphasise that it must be transboundary in nature. Effective protection, monitoring, and sustainable management of the Upper Zambezi must involve both Angola and Zambia. In particular, Angola's national water authorities, such as the Ministry of Energy and Water (MINEA) and its affiliated agencies, could play a central role in these efforts and collaborate with the Zambia Water Resources Management Authority. |

| Specific Comments | Authors response |
|---|---|
| l. 43. The river first traverses Southward over a very significant length. This is important to note as over that length also the ecology/climatology changes from equatorial forests with tropical climate to semi-arid Savannah. | We have revised the sentence to more accurately reflect the initial course of the Zambezi River, which first flows northward, then westward into Angola, before turning south and eventually eastward. We have also incorporated the suggestion regarding the ecological and climatic transitions - the mesic through arid savanna biomes - that occur along the river's southward course. These have been adjusted for this section of the introduction in the revised manuscript. |
| Table 2. The line on empirical discharge suggests that *in-situ* ADCP measurements would result in "mean discharge", which suggests mean in time. But that is not true. It provides only instantaneous discharge at the moment of the measurement. Such a measurement can never be used to conclude on accumulated contributions of a river tributary. See my main comments. | We agree that ADCP measurements represent instantaneous discharge at the time of sampling, not time-averaged or mean discharge over a longer period. Our original wording in Table 2 was indeed misleading in this regard. We have revised the relevant text in Table 2 to clarify that the discharge values reported are point-in-time (instantaneous) estimates, and that they should not be interpreted as representing mean or accumulated flow contributions of the tributaries over time. On line 289 in the original manuscript we state that: "As our ADCP readings provide a single discharge measurement at a point in time…"

On Table 2. The text has been altered to: "Point-in-time Discharge "(m3.s-1) and acquisition date"

However, we also note that (to our knowledge) these are the only direct discharge measurements available for these specific tributaries. Given the extreme data scarcity in the region, we consider these empirical observations to be valuable. While we acknowledge their limitations, we believe they can still offer insightful, even if provisional, information on relative flow magnitudes at the time of sampling.

We have updated the manuscript to reflect this dual perspective: (1) being cautious not to overinterpret these values as representative of long-term behaviour, but also (2) justifying their inclusion as the best available empirical evidence, which can serve as a starting point for future hydrological monitoring and analysis. |
| l. 290 onwards. How did you combine sampled ADCP observations with gauge records and the modeled records? What conclusions can you draw from that? This is quite important and a strong weakness in the paper. For instance, if you would take a end-of-dry-season low flow observation in the Zambezi main stem just below the Barotse floodplains, that will be relatively high (after dividing over the upstream area) compared to the low flow observations in faster responding systems such as the Luangwa. However, this does not say anything about the longer term contributions. Also, assuming you can use one single observation to scale or bias | We fully acknowledge and agree regarding the limitations of comparing single-point-in-time ADCP observations to longer-term gauge or modelled discharge records, especially given the seasonal variability in river response and potential biases in uncalibrated hydrological models. Its stated on line 290-292 in the original manuscript.

In our analysis, we did not use the instantaneous ADCP measurements to directly scale or bias-correct modelled flow data. Instead, our approach was to use these empirical measurements as contextual benchmarks, particularly in regions with no historical gauge data, to (1) qualitatively assess the plausibility of model estimates, and (2) highlight where discrepancies exist between model output and field observations. We clearly state in the manuscript that these are snapshot measurements, and we do not interpret them as proxies for long-term discharge or water yield.

Where possible, we compared our observations to nearby long-term gauge records and modelled trends, not to derive scaling factors but to provide triangulation and identify data gaps. This was especially valuable for Angolan (Lungwebungu and Luena) tributaries, where existing discharge data are extremely sparse or non-existent. |

| | |
|---|---|
| correct modeled flows is not a viable approach as hydrological models, that are not fully calibrated, will have seasonally varying biases. This comment also leads to my main comments. | We have revised the relevant section of the manuscript to clarify this methodology and added language that cautions against overinterpreting these measurements, while also emphasising their value in providing the only available empirical evidence in these headwater areas.

We note that ADCP (or comparative modelled flow and gauge station) measurements from the Luangwa system, an example of a fast-responding catchment, as highlighted by the reviewer, were not included in our analysis, as the Luangwa lies outside the Upper Zambezi basin. Our interpretation of river flow was deliberately restricted to the Upper Zambezi, upstream of the Barotse floodplains, focusing on headwater contributions from Angola and Zambia. This decision was based on the well-documented difficulties in accurately modelling the hydrology of the Barotse floodplain system through to Victoria Falls, as discussed in the literature and referenced throughout our original manuscript. Thus, we believe our estimates to be cautious in nature, reflecting only the uppermost sections of the Zambezi River Basin. |
| l. 303. Gaps are filled with EO datasets, but from what is written underneath, it is not clear how you would end up with reasonably bias-free river discharge numbers. It only describes typically available terrain-derived variables. | We agree that the description of how Earth Observation (EO) datasets were used to address data gaps requires more clarity, particularly regarding how we approached potential bias in estimating river discharge.

To clarify: we did not attempt to produce absolute river discharge values from EO data alone. Instead, EO datasets (wetland inventory and surface water seasonality and occurrence) were used to provide independent, spatially extensive indicators of hydrological processes and patterns, especially in ungauged sub-basins. These datasets helped to infer relative hydrological behaviour across tributaries and contextualise the discharge estimates, (from gauge stations, modelled data and our empirical measurements) not correct them.

We acknowledge that without full calibration and validation using long-term in situ data, absolute discharge estimates based on EO-derived proxies will carry significant uncertainty. However, in data-sparse headwaters like the Angolan Highlands and Lufilian Arc, such proxies offer critical insight into basin-scale dynamics, particularly when combined with empirical observations.

We have revised the relevant section to more clearly explain the role of EO data as supporting indicators, not direct substitutes for calibrated discharge models, and to explicitly acknowledge the associated uncertainty and limitations. |
| L. 315 onwards: Modelled runoff by Akpoti et al and Collins et al. Likely the very gauge records the authors use to compare it against, have also been used to perform monthly bias-correction of the data, and likely the sole reason to utilize these data for this study is to estimate river flows in ungauged areas, where (trivially) no records were available to perform bias correction. Given the highly variable landscapes throughout the basin areas (also described by the authors themselves) it is also not likely | We appreciate the reviewer's detailed comment regarding the use of modelled runoff datasets in heterogeneous and data-sparse basins. However, we would like to clarify that we did not use the Akpoti or Collins modelled runoff datasets in our core analysis. Rather, we compared these datasets to empirical discharge measurements (from the ADCP) and available historical gauge records, and found substantial discrepancies, both in terms of discharge magnitude and seasonality. These datasets were not sufficiently consistent with observed conditions to be considered reliable for our purposes. In response, we chose instead to present these datasets only to illustrate the broader challenge of estimating river discharge across the Upper Zambezi basin, particularly in ungauged areas such as the Angolan Highlands. This discrepancy has also been noted in previous literature and underscores the critical need for more empirical data in this region, a main point of our manuscript. |

| | |
|---|---|
| that the bias follows a similar pattern (e.g. the Zambian upper Zambezi is characterized by thick Kalahari sands, while the Angolan water tower does not have large alluvial aquifers). Furthermore, especially in semi-arid parts of the investigated basins, a small error in the modeled water balance will result in a very large error in the runoff and discharge, as discharge is a non-linearly responding residual of the hydrological processes. | Additionally, we respectfully disagree with the assertion that the Angolan Water Tower lacks significant subsurface storage. The region is also underlain by Kalahari Group sediments, which form extensive aquifers that play a crucial role in sustaining baseflow and supporting minerotrophic peatlands. These characteristics are central to the Angolan Highlands functioning as a hydrological water tower, not solely due to rainfall input, but also due to storage and delayed release mechanisms.

We have updated the manuscript to better clarify these points and ensure our interpretation and data usage are transparent. We also include a map in the supplementary material showing the distribution of Kalahari sediments across southern Africa, highlighting their expanse into the Angolan Highlands. |
| l. 335 "Given the limited availability of comprehensive water chemistry data across the Zambezi Basin, we have incorporated a wetland and surface water inventory for each HydroSHEDS Level 04 sub-basin to complement existing data and serve as a supporting proxy indicator of overall river and ecosystem health." What do you mean by a "wetland and surface water inventory". How do you use this to characterize ecosystem health? | Thank you for this important question. In the manuscript, our reference to a "wetland and surface water inventory" refers specifically to the use of satellite-derived surface water data from the Joint Research Centre (JRC) Global Surface Water Dataset, particularly the Occurrence and Seasonality layers. We applied this dataset across each HydroSHEDS Level 04 sub-basin to characterise patterns of surface water presence and persistence over time.

These metrics provide insights into the extent, frequency, and variability of surface water, important indicators for floodplain/ wetland dynamics and ecosystem function. In regions such as the Upper Zambezi and Angolan Highlands, which remain largely undammed and hydrologically unmodified, these surface water dynamics are shaped by natural hydrological processes. In contrast, downstream areas with large dams (e.g., Kariba and Cahora Bassa) exhibit artificially regulated surface water patterns that diverge significantly from their natural state.

Our rationale for including this analysis is that natural surface water dynamics are critical for sustaining floodplain ecosystems, biodiversity, water quality, and the overall resilience of riverine habitats. Therefore, we propose that consistent, naturally fluctuating surface water regimes, such as those observed in the undammed headwaters, can serve as indicators of ecosystem health. These indicators become especially valuable in data-scarce environments where long-term ecological monitoring is limited.

To clarify this point, we have revised the manuscript language to more explicitly describe the source of the data, our methodology, and its role in assessing ecosystem health. |
| l. 400. How were the errors of the river length calculated? | Thank you for this query, we appreciate the opportunity to clarify this. The river length error was calculated by comparing the computed river lengths derived from the ESRI Trace Downstream tool to our empirically measured expedition tracks, which serve as the ground-truth river paths.

The error percentage was then determined by calculating the relative difference in length between the ESRI modelled Length (EL) and the Expedition Track length (TL) for the river sections we traversed, and this detail has been provided in the relevant section. This comparison allows us to quantify the discrepancy between modelled hydrographic datasets and direct field observations, highlighting the potential inaccuracy of terrain-derived models, as demonstrated by our results. |

| | |
|---|---|
| l. 435. "flow accumulation pixels", please convert this into a metric unit such as km2. | Thank you for this helpful suggestion. We now present both the number of flow accumulation pixels and the corresponding area of accumulating pixels in km2. Flow accumulation values are derived from the HydroSHEDS 15 arc-second (~500 m at the equator) resolution DEM. In ArcGIS Pro, we calculated the area of each pixel in this region to be 0.198 km2. Including both the pixel count and its km2 equivalent allows us to retain the original analytical precision while also providing a more intuitive physical understanding of drainage areas. These data have been added to Table 4. |
| l. 441. "Sananga" should be "Senanga". | Thank you for this, we have corrected "Sananga" to "Senanga". |
| l. 458 – 466. The precision of the flow estimates and percentages contributions is unnecessarily and unrealistically high. Furthermore, it is unclear what can be drawn from the relative contributions during just one snap shot. | We appreciate the reviewer's concern regarding the precision and interpretation of our flow estimates. The reported precision reflects the variation observed during the ADCP measurements, where at least four runs were conducted at each site to calculate a mean discharge over the sampling period, not a single instantaneous measurement. While these still represent snapshots in time, the repeated runs improve the reliability of the flow estimates during sampling. We acknowledge the limitations inherent to such data, and have clarified this in the manuscript. Additionally, we emphasise that these measurements are used alongside longer-term datasets to provide context, rather than as standalone definitive values. |
| l. 490-506. Also a lot of very precise numbers. It is also not clear to me why rough estimates of percentage-wise contributions would have to rely on single snapshot ADCP observations. A much more logical proxy would be the catchment area multiplied by long-term runoff estimates using the Budyko curve. One snapshot just does not make sense as no seasonal variability or seasonal accumulation can be derived at all. | We appreciate the reviewer's concern regarding the precision and interpretation of our flow estimates. The reported precision reflects the variation observed during the ADCP measurements. We mentioned in the original manuscript, table 2, that we conducted four ADCP transects to calculate variance, and any site with high variance (coefficient of variation > 0.5) was resampled to ensure data quality.

We report flow values with two decimal places as per scientific standards. For smaller flows such as 1.55 m3/s at the source in Zambia, this level of precision is necessary because rounding can obscure meaningful differences. For larger flows, rounding has less impact on interpretation.

While these still represent snapshots in time, the repeated runs improve the reliability of the flow estimates during sampling. We acknowledge the limitations inherent to such data and have clarified this in the manuscript. Additionally, we emphasise that these measurements are used alongside long-term datasets to provide context, rather than as standalone definitive values. |
| Table 8. The description "Seasonality: Number of months water is present within a calendar year, average 1984 – 2021." is not clear to me. The table shows percentages for each individual month, not yearly aggregates. | Thank you for your observation. We acknowledge that the original description lacked clarity. Table 8 does not show the percentage of water presence for each calendar month. Instead, it presents a normalised distribution of pixels that experience inundation, categorised by the number of months per year in which each pixel is typically inundated. That is, it shows what proportion of inundated pixels are inundated for 1 month, 2 months, ..., up to 12 months annually, based on the average for 1984–2021.

The table title has been changed to: Distribution of inundated pixels by the number of months water is present per year, averaged over 1984–2021. Values represent the percentage of inundated pixels that are typically inundated for 1 to 12 months annually. |

| | |
|---|---|
| Fig 7. Is subplot b not simply a differently scaled version of subplot a? If not please explain what both maps exactly mean and why both are required for your analysis. | Thank you for the comment. Subplots (a) and (b) in Figure 7 represent two distinct but complementary metrics from the JRC dataset and are not merely differently scaled versions of the same data. Panel (a) shows average water occurrence, the percentage of time water has been present at each pixel from 1984 to 2021. Panel (b) shows seasonality, indicating the number of months per year water is typically present at each pixel.

Together, these metrics help characterise surface water dynamics. Importantly, the figure illustrates the contrast between the natural, undammed Upper Zambezi, where water presence follows natural seasonal patterns, and the Zambezi river downstream of Victoria Falls, where water pooling and damming (besides natural lakes) have altered the natural surface water occurrence and seasonality.

Including both metrics enhances understanding of how reservoirs affect floodplain hydrology and ecosystem health, a main focus of our manuscript. |
| l. 626 – 634. I do not see the scientific importance of the "finding" that the Lungwebungu is longer than the length from Lukulu to the source. It is merely a geographical characteristic. | We respectfully disagree with the suggestion that the length of the Lungwebungu tributary is merely a geographical characteristic without scientific importance. By hydrological convention, the source of a river is defined as its longest tributary, making this a fundamental criterion in identifying the source of the Zambezi River.

It is widely accepted in the geomorphological and hydrological literature that primary channel length is a fundamentally important variable/factor influencing earth surface processes.

While we acknowledge that other factors may contribute to defining a river's source (Table 1), length remains a primary and widely accepted metric. In our study, we also included additional defining criteria that collectively provide evidence suggesting that the Zambezi's source lies within Angola, rather than Zambia. |
| l. 643 – 644 "Although the Angolan portion covers only 14% more area, it contributes approximately 45.58% more flow at the Lukulu confluence than the Zambian portion." How is this concluded? If that comes from the snapshot ADCP observations, then the conclusion is not valid, (see main comment) as you would require multi-year time series to properly assess this. | We appreciate the reviewer's concern regarding the conclusion on flow contributions from the Angolan and Zambian portions of the basin. This conclusion is based on a combination of historical gauge station data spanning multiple years, as well as our empirical discharge measurements, which we have clearly stated in the original manuscript. While we acknowledge that the snapshot ADCP observations alone cannot fully capture long-term variability, the integration with multi-year gauge records provides initial quantitative evidence that the Angolan portion likely contributes significantly more flow at the Lukulu confluence, upstream of the Barotse wetlands.

We emphasise that this represents an important first step toward better understanding these contributions. It highlights priorities, and so motivates for continued and expanded monitoring to capture seasonal and interannual variability more comprehensively. We have clarified this point in the revised manuscript to ensure the limitations and context of the data are transparent. |
| l. 659 – 666. Not really something that belongs in the discussion, more logical for the introduction. | Thank you for this helpful suggestion. We agree that the description of river sources and associated wetland systems provides important contextual background and is better suited for the Introduction. We have moved this paragraph accordingly and made minor edits to improve its integration into the introductory section. |

| l. 669 – 677, also not a discussion based on this work, merely some text about the basin from existing references. It does not belong in this section to my mind. | Thank you for this valuable feedback. We agree that this paragraph primarily provides contextual information and does not directly engage with our findings. We have therefore integrated aspects of this paragraph into the Study Area section, where it better supports the description of our study area. |
|---|---|
| l. 679 – 688. Also this contains no discussion points based on this work. | We appreciate the comment and understand the concern regarding the relevance of this section to the discussion. However, we respectfully disagree, as this paragraph synthesises historical hydrological data from both the Okavango and Zambezi Basins to underscore a key finding of our study: the shared influence of the Angolan Highlands Water Tower (AHWT) on these transboundary river systems. While the long-term records cited are from existing literature, our contribution lies in highlighting their overlooked connection and demonstrating how precipitation variability in the AHWT region affects both basins simultaneously. This connection directly supports our broader argument about the critical hydrological and ecological importance of the AHWT, which is a central focus of the manuscript. |
| l. 708 "Its size is the reason for over-estimations of modelled discharge". Why would the size result in an overestimation? It is more likely poor representation of semi-arid hydrological processes in the used models or biased rainfall. | Thank you for this insightful comment. We agree that overestimation of discharge in the Cuando River is not solely a function of basin size but is also linked to how flow accumulation-based models like those from Akpoti et al. and Collins et al. operate. These modelled datasets use flow accumulation as a proxy for upstream contributing area, which can be effective in many contexts. However, in semi-arid and floodplain-dominated systems such as the Cuando, flow accumulation alone does not adequately capture key hydrological processes such as high evaporation, infiltration, and extended water residence times; we mentioned this in the original manuscript, lines 718-720.

The Cuando basin is indeed large, and this contributes to high flow accumulation values. However, this does not translate into proportionally high discharge, as demonstrated by long-term observational data from the Kongola gauge station. Here, we observe relatively low flows throughout the year, with only minor seasonal variation. In contrast, the modelled data significantly overestimate both magnitude and seasonality of discharge, with peaks out of season with observed data. This discrepancy underscores the difficulty of applying standard flow accumulation based models to regions with complex floodplain hydrology.

Therefore, our manuscript highlights that it is not merely the basin's physical size causing overestimation, but rather the model assumptions that fail to represent semi-arid hydrological processes, particularly those involving storage, evaporation, and attenuation in floodplains. We have clarified this discussion in the revised manuscript and emphasised the need for caution when using such datasets in the Cuando and similar basins. |
| l. 731. "pivotal—yet previously overlooked—implications for transboundary river management". This sounds very strong again, as if no-one ever thought about the fact that a large part of the Zambezi's water stems from Angola. | We thank the reviewer for this observation. We are not claiming that no one ever knew that water comes from Angola, but rather that its critical role has been underemphasised, and this omission has real consequences for management and policy.

To our knowledge, there has been no prior study or report that quantitatively assesses the Angolan contribution to the Zambezi River flow. While it is generally recognised that Angola forms part of the upper catchment, our study represents the first attempt to provide empirical quantification and analysis of its specific contribution. |

We emphasise that this estimation is an initial step, and we strongly advocate for the establishment of long-term monitoring stations in Angola, as well as maintenance of existing stations in Zambia, to verify and improve these assessments. This is a crucial need for effective transboundary water management in the basin. We have clarified these points in the revised manuscript to better contextualise our findings and their implications.

---

## Author Comment (AC2)

**Author's response to Reviewer 2 comments**

EGUSPHERE-2025-837 | Research article

**Manuscript Title: Assessment of source regions of the Zambezi River: implications for regional water security.**

The authors would like to thank the anonymous reviewer and the editor (Professor Hubert H.G. Savenije) for their time and effort in providing their expert views and comments on the manuscript during the review process.

As per the journal's instructions, we are currently in the final response phase and are providing detailed replies to all referee and community comments. A revised manuscript has not yet been submitted, as this is not requested at this stage of the review process. Where relevant, we describe changes to the manuscript by referring to specific sections or content rather than line numbers or tracked edits, since the revised manuscript is not provided at this stage.

We have made every effort to address the points raised, as detailed below.

**Reviewer 2: Citation: https://doi.org/10.5194/egusphere-2025-837-RC2**

| Overall comments | Authors response |
| --- | --- |
| I have reviewed the submission by Lourenco et al, on the assessment of source regions of the Zambezi River: Implications for regional water security. The manuscript investigates source regions of the Zambezi water towers (Luffician Arc and Angolan Highlands). These regions are data scarce and critical for water resources management at transboundary level. The following should be addressed to make it suitable for an international journal like HESS: | We thank the reviewer for their careful reading and thoughtful suggestions, which have helped us strengthen and clarify the manuscript. We address each of the points raised below.

We hope that these comments and revisions make the manuscript's contributions more compelling and clear for a broader international readership. |

| Specific comments | Authors response |
| --- | --- |
| 1. Novelty - What is the scientific knowledge that this study is bringing forward/advancing. For the people in the Zambezi Basin, this is clear but the study must appreciated by a broader audience. | We appreciate this important question and agree that we needed to more clearly articulate the novelty of our contribution for an international audience. While local and regional researchers may be aware of the Angolan Highlands' importance to the Zambezi system, there has been no integrated assessment of the hydrological, geomorphological, and water quality contributions of these headwaters across multiple disciplines and data sources.

The opening paragraph of the Discussion summarizes the salient findings of this study. And here we present their wider context and relevance to the readership of this journal.

To our knowledge, this was the first study to:

1. Conduct GPS-tracked ground expeditions to map the Zambezi River from both Zambian and Angolan sources. The hydrological and geomorphological knowledge of the headwaters was aided by the use of these transects to measure and compare river lengths.
2. Specifically in Angola, where no previous discharge data has been published or reported, we present new ADCP-based flow measurements across a number of important tributaries. For many of these sites, these are the only field-based discharge data available. By merging this flow data with previous gauge station data, the Angolan and Zambian contribution to the Zambezi (upstream of the Barotse wetland) was estimated. |

| | 3. We examined two of the most recent EO-based modelled discharge products for the Upper Zambezi sub-basins (Collins et al., 2024, and Akpoti et al., 2024) and found inconsistencies and limits in comparison to our ADCP data and historical gauge station data. |
|---|---|
| | 4. Provide an inventory of wetlands and the occurrence and seasonality of surface water, including assessments of water chemistry and quality, emphasising the Upper Zambezi's natural (undammed) hydrological behaviour, especially in Angola, which has received less attention in the literature than Zambia's more extensively researched wetland areas. |
| | In response, we have revised the abstract, introduction, and discussion to better highlight the manuscript's key contributions. |
| 2. Unvalidated statements: In sections of the paper, the authors highlight two things which I feel are unvalidated: (a) reduced elevated conductivity, salinity, TDS are likely introduced by mining. Several assessments conducted in this catchment do NOT show this. In any case, can you explain how mining is reducing the EC? (b) Angola hosts primary aquifers of the Zambezi - again this is difficult to validate based on the data available in this manuscript. | We note the need for caution and clarification regarding these statements. We have clarified these points in the revised manuscript to better contextualise our findings and their implications. |
| | We are satisfied that our data provides a direct and temporally aligned comparison regarding the elevated conductivity (rather than reduced EC as noted by the reviewer) from the Kabompo River in comparison to the Zambezi and Lungwebungu River systems. Our findings align with broader concerns about anthropogenic pressures in the Kabompo/Copperbelt region, which are well-documented. Our intent was not to present this comparison as a definitive, seasonally controlled analysis, but rather as an initial observational contrast that raises important questions about localised impacts of land use, particularly mining, on water quality in Zambia. Note that the Wilderness Project is currently conducting (having started on 3 June 2025) a river expedition specifically along the Kabompo River in Zambia, with the intention of reporting further details regarding potential water chemistry concerns on this major tributary. |
| | Regarding the AHWT as the primary aquifer of the Zambezi, we have made changes to the manuscript that softens this stance, in line with the need for caution. We acknowledge that this work is foundational rather than definitive, serving as a call for further research into the entire Upper Zambezi and, crucially, for long-term monitoring. We advocate for the establishment of gauging stations, particularly in Angola, and the maintenance of existing stations in Zambia, to build upon this first step in quantifying the hydrology (and primary aquifer) of this critical region. |
| 3. Flow accumulation and river discharge: Measurements of ADCP need to be clear if they are mean or one-time measurements. In some section, specific dates are placed, whereas other not placed. | Thank you for the comment, we have clarified that all ADCP measurements were single-time observations taken during specific expedition dates. Their mean value indicates the mean over the sampling effort and does not represent a mean daily/ monthly flow, we appreciate that this can be confusing to our audience. The dates had been added in all relevant sections and figure captions in the original draft. We also now more clearly state the limitations of this snapshot approach and that the results should be seen as indicative rather than representative of long-term flow. |
| 4. Some sections of the introduction can be reduced and benefit from a refined perspectives on headwater analysis/ literature review. For me section 1.1 and 1 should be combined. | We agree that the introduction can be made more concise and focused. In response, we have revised and streamlined the content, combining Sections 1 and 1.1 into a single, more cohesive introduction. In doing so, we have refined the discussion of headwater analysis and related literature to better frame the context and motivation for our study. |

---

## Author Response (AR2)

**Author's response to the Editor following revisions**

EGUSPHERE-2025-837 | Research article

**Manuscript Title: Assessment of source regions of the Zambezi River: implications for regional water security.**

The authors would like to thank the anonymous reviewers and the editor (Professor Hubert H.G. Savenije) for their time and effort in providing their expert views and comments on the manuscript during the review process.

| Public Justification | Authors response |
|---|---|
| Dear Authors,
Thank you for extensively replying to the comments of the referees and for modifying the paper in a way that most of the criticism is neutralized.
I am willing to accept the paper. Although the two reviewers pointed out that the innovation in the paper is limited, with which I agree, I think that the hydrological community can benefit very much from the detailed field survey that the authors have carried out and the extensive analysis and reporting of their results. They authors provided a valuable hydrographic, geographic and ecological background and literature study of the Zambezi tributaries and, what is most important, provided an insight into a part of the Zambezi that was not surveyed in so much detail before. Even though it was carried out within a short window of time, which does not say much about the longer term or average conditions, I do think it provides a valuable snapshot of the hydrological condition of particularly the Angolan part of the catchment.
The paper is well written and well documented, and it has taken the comments of the referees at heart. The paper present a unique data set of detailed and high density observations of river discharge and water quality in previously ungauged and unsampled Zambezi tributaries, which are a welcome addition to the hydrological knowledge of the Zambezi.

I still have a few minor technical issues to address.
1. I think that there is not much justification to add both Fig7a and Fig7b. The are almost identical and their difference is not discussed. So maybe stick with the seasonality graph and remove the occurrence map.
2. in line 635 write 97 km3/yr, because it is a flux and not a stock. | Dear Professor Hubert H.G. Savenije

Thank you very much for your thoughtful comments and for accepting our manuscript for publication in EGUSPHERE *Hydrology and Earth System Sciences.* We are sincerely grateful for your careful review and for recognising the value of our fieldwork and reporting on the headwaters of the Zambezi.

We particularly appreciate your public statement, and the decision to select the paper as a highlight, which thoughtfully acknowledges the significance of documenting this largely ungauged and under-studied region. It is encouraging to know that the insights and dataset we provide are seen as a meaningful contribution to the hydrological understanding of the Zambezi basin, especially within the Angolan catchment.

Regarding your remaining technical comments:

We agree that Figures 7a and 7b are very similar and that their differences are not essential to the current discussion. We have removed the occurrence map and retain the seasonality map, as suggested.

We have revised the unit in line 632 to "97 km³/yr" to reflect the correct expression of a flux.

Thank you once again for your support and guidance throughout the review process.

Dr. Mauro Lourenco (on behalf of all co-authors) |